# Faster Algorithms for User-Level Private Stochastic Convex Optimization[*]

**Andrew Lowy**
Wisconsin Institute for Discovery
University of Wisconsin-Madison
alowy@wisc.edu

**Daogao Liu**
Department of Computer Science
University of Washington
liudaogao@gmail.com

**Hilal Asi**
Apple Machine Learning Research
hilal.asi94@gmail.com

## Abstract

We study private stochastic convex optimization (SCO) under user-level differential privacy (DP) constraints. In this setting, there are $n$ users (e.g., cell phones), each possessing $m$ data items (e.g., text messages), and we need to protect the privacy of each user's entire collection of data items. Existing algorithms for user-level DP SCO are impractical in many large-scale machine learning scenarios because: (i) they make restrictive assumptions on the smoothness parameter of the loss function and require the number of users to grow polynomially with the dimension of the parameter space; or (ii) they are prohibitively slow, requiring at least $(mn)^{3/2}$ gradient computations for smooth losses and $(mn)^3$ computations for non-smooth losses. To address these limitations, we provide novel user-level DP algorithms with state-of-the-art excess risk and runtime guarantees, without stringent assumptions. First, we develop a *linear-time* algorithm with state-of-the-art excess risk (for a non-trivial linear-time algorithm) under a mild smoothness assumption. Our second algorithm applies to arbitrary smooth losses and achieves *optimal excess risk* in $\approx (mn)^{9/8}$ gradient computations. Third, for *non-smooth* loss functions, we obtain *optimal excess risk* in $n^{11/8}m^{5/4}$ gradient computations. Moreover, our algorithms do not require the number of users to grow polynomially with the dimension.

## 1 Introduction

The increasing ubiquity of machine learning (ML) systems in industry and society has sparked serious concerns about the privacy of the personal data used to train these systems. Much work has shown that ML models may violate individuals' privacy by leaking their sensitive training data [SSSS17, LLL+24a, LLL+24b]. For instance, large language models (LLMs) are vulnerable to black-box attacks that extract individual training examples [CTW+21]. *Differential privacy* (DP) [DMNS06] prevents ML models from leaking their training data.

The classical definition of differential privacy—*item-level differential privacy* [DMNS06]—is ill-suited for many modern applications. Item-level DP ensures that the inclusion or exclusion of any *one training example* has a negligible impact on the model's outputs. *If each person (a.k.a. user) contributes only one piece of training data*, then item-level DP provides a strong guarantee that each user's data cannot be leaked. However, in many modern ML applications, such as training LLMs on users' data in federated learning, each user contributes a large number of training examples [XZ24].

---

[*] Authors are listed reverse alphabetically.

38th Conference on Neural Information Processing Systems (NeurIPS 2024).

In such scenarios, the privacy protection that item-level DP provides for each user is insufficiently weak.

*User-level differential privacy* is a stronger privacy notion that addresses the above shortcoming of item-level DP. Informally, user-level DP ensures that the inclusion or exclusion of any *one user's entire training data* ($m$ samples) has a negligible impact on the model's outputs. Thus, user-level DP provides a strong guarantee that no user's data can be leaked, even when users contribute many training examples.

A fundamental problem in (private) machine learning is *stochastic convex optimization* (SCO): given a data set $\mathcal{D} = (Z_1, \ldots, Z_n)$ from $n$ i.i.d. users, each possessing $m$ i.i.d. samples from an unknown distribution $Z_i \sim P^m$ our goal is to approximately minimize the expected population loss

$$F(x) := \mathbb{E}_{z \sim P}[f(x, z)].$$

Here, $f : \mathcal{X} \times \mathcal{Z} \to \mathbb{R}$ is a loss function (e.g., cross-entropy loss), $\mathcal{X} \subset \mathbb{R}^d$ is the parameter domain, and $\mathcal{Z}$ is the data universe. We require that the output of the optimization algorithm $\mathcal{A} : \mathcal{Z}^{mn} \to \mathcal{X}$ satisfies user-level DP (Definition 1.3). We measure the accuracy of $\mathcal{A}$ by its *excess (population) risk*

$$\mathbb{E}F(\mathcal{A}(\mathcal{D})) - F^* := \mathbb{E}_{\mathcal{A}, \mathcal{D} \sim P^{nm}} F(\mathcal{A}(\mathcal{D})) - \min_{x \in \mathcal{X}} F(x).$$

Given the practical importance of user-level DP SCO, it is unsurprising that many prior works have studied this problem. The work of [LSA+21] initiated this line of work, and provided an excess risk lower bound of $\Omega(1/\sqrt{nm} + \sqrt{d}/(\varepsilon n \sqrt{m}))$, where $\varepsilon$ is the privacy parameter. However, their upper bound was suboptimal and required strong assumptions. The work of [BS23] gave an algorithm that achieves optimal risk for $\beta$-smooth losses with $\beta < (n/\sqrt{md} \wedge n^{3/2}/(d\sqrt{m}))$, provided that $n \geq \sqrt{d}/\varepsilon$ and $m \leq \max(\sqrt{d}, n\varepsilon^2/\sqrt{d})$. *These assumptions are restrictive* in large-scale applications with a large number of examples per user $m$ or when the number of model parameters $d$ is large. For example, in deep learning, we often have $d \gg n$ and an enormous smoothness parameter $\beta \gg 1$. Moreover, their algorithm requires $mn^{3/2}$ gradient evaluations, making it slow when the number of users $n$ is large.[2] The work of [GKK+23] gave another user-level DP algorithm that only requires $n \geq \log(d)/\varepsilon$, but unfortunately their algorithm does not run in polynomial-time.

To address the deficiencies of previous works on user-level DP SCO, the recent work [AL24] provided an algorithm that achieves optimal excess risk in polynomial-time, while also only requiring $n \geq \log(md)/\varepsilon$ users. Moreover, their algorithm also works for non-smooth losses. The drawback of [AL24] is that it is even *slower* than the algorithm of [BS23]: for $\beta$-smooth losses, their algorithm requires $\beta \cdot (nm)^{3/2}$ gradient evaluations; for non-smooth losses, their algorithm requires $(nm)^3$ evaluations.

Evidently, *the runtime requirements and parameter restrictions of existing algorithms for user-level DP SCO are prohibitive in many important ML applications*. Thus, an important question is:

> **Question 1.** Can we develop *faster* user-level DP algorithms that achieve *optimal* excess risk *without restrictive assumptions*?

**Contribution 1.** We give a positive answer to Question 1, providing a novel algorithm that achieves optimal excess risk using $\max\{\beta^{1/4}(nm)^{9/8}, \beta^{1/2}n^{1/4}m^{5/4}\}$ gradient computations for $\beta$-smooth loss functions, with any $\beta < \infty$ (Theorem 3.2). For non-smooth loss functions, our algorithm achieves optimal excess risk using $n^{11/8}m^{5/4}$ gradient evaluations for non-smooth loss functions (Theorem 4.1). *Our runtime bounds dominate those of all prior works* in every applicable parameter regime, by polynomial factors in $n, m,$ and $d$. Moreover, our results only require $n^{1-o(1)} \geq \log(d)/\varepsilon$ users. See Table 1 for a comparison of our results vs. prior works. For example, *for non-smooth loss functions, our optimal algorithm is faster than the previous state-of-the-art [AL24] by a multiplicative factor of $n^{13/8}m^{7/4}$. For smooth loss functions, our optimal algorithm is faster than [AL24] by a factor of $(nm)^{3/8}\beta^{3/4}$ (in the typical parameter regime when $n^7 \geq m$).

---

[2]In the introduction, whenever $\varepsilon$ does not appear, we are assuming $\varepsilon = 1$ to ease readability. For runtime bounds, we also assume $n = d$ to further simplify.

| Loss Function | Reference | Gradient complexity | Assumptions |
|---|---|---|---|
| **$\beta$-Smooth** | [BS23] | $mn^{3/2}$ | $\beta \leq \sqrt{n/m}$ & $m \leq \sqrt{d} \leq n$ |
| | [AL24] | $\beta \cdot (mn)^{3/2}$ | None |
| | Our Algorithm 3 | $\beta^{1/4} \cdot (mn)^{9/8} + \beta^{1/2} n^{1/4} m^{5/4}$ | None |
| **Non-Smooth** | [AL24] | $(mn)^3$ | None |
| | Our Algorithm 3 (smoothed) | $n^{11/8} m^{5/4}$ | None |

Figure 1: Optimal algorithms for user-level DP SCO. We omit logarithms, fix $L = R = 1 = \varepsilon$ and $n = d$.

**Linear-Time Algorithms**   The "holy grail" of DP SCO is a *linear-time* algorithm with optimal excess risk, which is unimprovable both in terms of runtime and accuracy. In the *item-level* DP setting, such algorithms are known to exist for smooth loss functions [FKT20, ZTOH22]. [AL24] posed an interesting open question: is there a *user-level* DP algorithm that achieves optimal excess risk in linear time for smooth functions? For our second contribution, we make progress towards answering this question.

Existing techniques for user-level DP SCO are not well-suited for linear-time algorithms. Indeed, the only prior non-trivial linear-time algorithm is the user-level LDP algorithm of [BS23, Algorithm 5].[3] Their algorithm can achieve excess risk $\approx 1/\sqrt{nm\varepsilon} + \sqrt{d}/(\sqrt{nm}\varepsilon)$. Unfortunately, however, their algorithm requires a very stringent assumption on the smoothness parameter $\beta < \sqrt{n^3/(md^3)}$, which is unlikely to hold for large-scale ML problems. Further, the result of [BS23] requires the number of users queried in each round to grow polynomially with the dimension $d$, and it assumes $m < d < n$. *These assumptions severely limit the applicability of [BS23, Algorithm 5] in practical ML scenarios.* This leads us to:

> **Question 2.** Can we develop a *linear-time* user-level DP algorithm with state-of-the-art excess risk, *without restrictive assumptions*?

**Contribution 2.**   We answer Question 2 affirmatively in Theorem 2.1: under a very mild requirement on the smoothness parameter $\beta < \sqrt{nmd}$, our novel linear-time algorithm achieves excess risk of $\approx 1/\sqrt{nm\varepsilon} + \sqrt{d}/(\sqrt{nm}\varepsilon)$. Moreover, our algorithm does not require the number of users to grow polynomially in the dimension $d$, and our result holds for any values of $m, d$, and $n$. Thus, our algorithm has excess risk matching that of [BS23], but is much more widely applicable.

### 1.1   Techniques

We develop novel techniques and algorithms to achieve new state-of-the-art results in user-level DP SCO. Before discussing our techniques, let us review the key ideas from prior works that we build on.

The goal of prior works [BS23, AL24] was to develop user-level analogs of DP-SGD [BFTT19], which is optimal in the item-level setting. To do so, they observed that each user $i$'s gradient $\frac{1}{m} \sum_{j=1}^m \nabla f(x, Z_{i,j})$ lies in a ball of radius $\approx 1/\sqrt{m}$ around the population gradient $\nabla F(x)$ with high probability, if the data is i.i.d ($Z_i \sim P^m$). Consequently, if the data is i.i.d., then replacing one user $Z_i \in \mathcal{D}$ by another user $Z_i' \in \mathcal{D}'$ will not change the empirical gradient $\nabla F_{\mathcal{D}}(x)$ by too much: $\|\nabla F_{\mathcal{D}}(x) - \nabla F_{\mathcal{D}'}(x)\| \lesssim 1/(n\sqrt{m})$ with high probability. Thus, one would hope for a method to privatize $\nabla F_{\mathcal{D}}(x)$ by adding noise that scales with $1/(n\sqrt{m})$—rather than $1/n$— which would allow for optimal excess risk. [AL24] devised such a method, which was inspired by FriendlyCore [TCK+22]. Their method privately detects and removes "outlier" user gradients, and then adds noise to the average of the "inlier" user gradients. This outlier-removal procedure ensures privacy with noise scaling with $1/(n\sqrt{m})$, provided $n \gtrsim 1/\varepsilon$. Moreover, when the data is i.i.d., no

---

[3]It is trivial to achieve excess risk $\approx 1/\sqrt{nm} + \sqrt{d}/(\varepsilon n)$ with $(\varepsilon, \delta)$-user-level, e.g. by applying *group privacy* to an optimal item-level DP algorithm such as [FKT20]. The error due to privacy in this bound does not decrease with $m$.

outliers will be removed with high probability, leading to a nearly unbiased estimator of the empirical gradient.

Our algorithms apply variations of the outlier-removal idea of [AL24] in novel ways.

Our linear-time Algorithm 1 takes a different approach to outlier removal, compared to prior works. Instead of removing outlier *gradients*, we aim to detect and remove outlier SGD *iterates*.[4] The high-level idea of our algorithm is to partition the $n$ users into $C \approx 1/\varepsilon$ groups, with each group containing $\approx n\varepsilon$ users. For each group of users, we run $T \approx mn\varepsilon$ steps of online SGD using the samples in this group and obtain the average iterate of each group: $\{\tilde{x}_j\}_{j=1}^C$. We then *privately identify and remove the outlier iterates* from $\{\tilde{x}_j\}_{j=1}^C$. In order to successfully do so, we need to argue that if we run online SGD independently on user $Z$ and user $Z'$ to obtain $\tilde{x}$ and $\tilde{x}'$ respectively, then $\|\tilde{x} - \tilde{x}'\| \lesssim \eta\sqrt{T}$ with high probability, where $\eta$ is the SGD step size. We prove such a stability bound in Lemma 2.3, which we hope will be of independent interest. By repeating the above process $\log(n)$ times and using iterative *localization* [FKT20], we obtain our state-of-the-art linear-time result.

Our second algorithm, Algorithm 3, builds on [AL24] in a different way. In Algorithm 3, we apply an outlier-removal procedure to users' gradients. However, unlike [AL24], we draw random *minibatches* of users in each iteration and apply outlier-removal to these minibatches. To make this procedure private while also achieving optimal excess risk, we combine *AboveThreshold* [DR14] with *privacy amplification by subsampling* [BBG18]. We then develop an *accelerated* [GL12] user-level DP algorithm that solves a carefully chosen sequence of regularized ERM problems, and applies localization in the spirit of [KLL21, AFKT21]. An obstacle that arises when we try to extend the ERM-based localization framework to the user-level DP setting is getting a tight bound on the variance of our minibatch stochastic gradient estimator that scales with $1/m$. We overcome this obstacle in Lemma 3.5, by appealing to the *stability of user-level DP* [BS23]. To handle non-smooth loss functions, we apply randomized smoothing to our accelerated algorithm.

## 1.2 Preliminaries

We consider loss functions $f : \mathcal{X} \times \mathcal{Z} \to \mathbb{R}$, where $\mathcal{X}$ is a convex parameter domain and $\mathcal{Z}$ is a data universe. Let $P$ be an unknown data distribution and $F(x) := \mathbb{E}_{z \sim P}[f(x, z)]$ be the population loss function. Denote $F^* := \min_{x \in \mathcal{X}} F(x)$. The SCO problem is $\min_{x \in \mathcal{X}} F(x)$. Let $\|\cdot\|$ denote the $\ell_2$ norm. $\Pi_{\mathcal{X}}(u) := \text{argmin}_{x \in \mathcal{X}} \|u - x\|^2$ denotes projection onto $\mathcal{X}$.

**Assumptions and Notation.** Function $g : \mathcal{X} \to \mathbb{R}$ is *L-Lipschitz* if $|g(x) - g(x')| \leq L\|x - x'\|_2$ for all $x, x' \in \mathcal{X}$. Function $g : \mathcal{X} \to \mathbb{R}$ is *$\beta$-smooth* if $g$ is differentiable and has $\beta$-Lipschitz gradient: $\|\nabla g(x) - \nabla g(x')\|_2 \leq \beta\|x - x'\|_2$. Function $g : \mathcal{X} \to \mathbb{R}$ is *$\mu$-strongly convex* if $g(\alpha x + (1 - \alpha)x') \leq \alpha g(x) + (1 - \alpha)g(x') - \frac{\alpha(1-\alpha)\mu}{2}\|x - x'\|^2$ for all $\alpha \in [0, 1]$ and all $x, x' \in \mathcal{X}$. If $\mu = 0$, we say $g$ is *convex*.

**Assumption 1.1.** *1. The convex set $\mathcal{X}$ is compact with $\|x - x'\| \leq R$ for all $x, x' \in \mathcal{X}$.*

*2. The loss function $f(\cdot, z)$ is L-Lipschitz and convex for all $z \in \mathcal{Z}$.*

In all of the paper *except for Section 4*, we will also assume:

**Assumption 1.2.** *The loss function $f(\cdot, z)$ is $\beta$-smooth for all $z \in \mathcal{Z}$.*

Denote $a \wedge b := \min(a, b)$. For functions $f$ and $g$ of input parameters $\theta$, we write $f \lesssim g$ if there is an absolute constant $C > 0$ such that $f(\theta) \leq Cg(\theta)$ for all permissible values of $\theta$. We use $\widetilde{O}$ to hide logarithmic factors. Write $a \leq \text{poly}(b)$ if there exists some large $J > 1$ for which $a \leq b^J$.

**Differential Privacy.**

**Definition 1.3** (User-Level Differential Privacy). Let $\varepsilon \geq 0$, $\delta \in [0, 1)$. Randomized algorithm $\mathcal{A} : \mathcal{Z}^{nm} \to \mathcal{X}$ is $(\varepsilon, \delta)$-*user-level differentially private* (DP) if for any two datasets $\mathcal{D} = (Z_1, \ldots, Z_n)$ and $\mathcal{D}' = (Z_1', \ldots, Z_n')$ that differ in one user's data (say $Z_i \neq Z_i'$ but $Z_j = Z_j'$ for $j \neq i$), we have

$$\mathbb{P}(\mathcal{A}(\mathcal{D}) \in S) \leq e^\varepsilon \mathbb{P}(\mathcal{A}(\mathcal{D}') \in S) + \delta,$$

---

[4]The reason that this innovation is necessary is discussed in the last paragraph of Section 2.

for all measurable subsets $S \subset \mathcal{X}$.

Definition 1.3 prevents any adversary from learning much more about an individual's data set than if that data had not been used for training. Appendix A contains the necessary background on DP.

### 1.3 Roadmap

We begin with our state-of-the-art linear-time algorithm in Section 2. In Section 3, we present our error-optimal algorithm with state-of-the-art runtime for smooth loss functions. Section 4 extends our fast optimal algorithm to non-smooth loss functions. We conclude in Section 5 with a discussion and guidance on future research directions stemming from our work.

## 2   A state-of-the-art linear-time algorithm for user-level DP SCO

In this section, we develop a new algorithm (Algorithm 1) for user-level DP SCO that runs in linear time and has state-of-the-art excess risk, without requiring any impractical assumptions. The algorithm can be seen as a user-level DP variation of the localized phased SGD of [FKT20]: we execute a sequence of SGD trajectories with geometrically decaying step sizes, shrinking both the expected distance to the population minimizer and the privacy noise over a logarithmic number of phases.

In each phase $i$, we first re-set algorithmic parameters and draw a disjoint set of $n_i$ users $D_i \subset \mathcal{D}$ (lines 4-5). We further partition $D_i$ into $C$ disjoint subsets $\{D_{i,j}\}_{j=1}^{C}$. For each $j \in [C]$, we pool together all of the $n_i m$ samples in $D_{i,j}$ and run one-pass online SGD on $D_{i,j}$ with initial point $x_{i-1}$ given to us from the previous phase. Next, in lines 10-20, we privately detect and remove "outliers" from $\{\tilde{x}_{i,j}\}_{j=1}^{C}$. That is, our goal is to privately select a subset $\mathcal{S}_i \subset \{\tilde{x}_{i,j}\}_{j=1}^{C}$, such that for any two points $\tilde{x}_{i,j}, \tilde{x}_{i,j'} \in \mathcal{S}_i$, $\|\tilde{x}_{i,j} - \tilde{x}_{i,j'}\| \leq \tau_i = \widetilde{O}(\eta_i L \sqrt{T_i})$. This will enable us to add noise scaling with $\tau_i$ in line 22, rather than with the much larger worst-case sensitivity (that scales linearly with $T_i$). In order to privately select such a subset $\mathcal{S}_i$, we first compute (and privatize) the *concentration score* for $\{\tilde{x}_{i,j}\}_{j=1}^{C}$ in line 10. A small concentration score indicates that outlier removal is doomed to fail and we must halt the algorithm to avoid breaching the privacy constraint. A large concentration score indicates that $\{\tilde{x}_{i,j}\}_{j=1}^{C}$ is nearly $\tau_i$-concentrated and we may proceed with outlier removal in lines 12-15.

**Theorem 2.1** (Privacy and utility of Algorithm 1 - Informal). *Let $\varepsilon \leq 10$, $n^{1-o(1)} \gtrsim \frac{\log(n/\delta)}{\varepsilon}$, $\beta \leq (L/R)\sqrt{dmn\varepsilon}$, and $m \lesssim poly(n)$. Then, Algorithm 1 is $(\varepsilon, \delta)$-user-level DP. Further,*

$$\mathbb{E}F(x_l) - F^* \leq LR \cdot \widetilde{O}\left(\frac{1}{\sqrt{nm\varepsilon}} + \frac{\sqrt{d\log(1/\delta)}}{\sqrt{n}\varepsilon\sqrt{m}}\right).$$

*The gradient complexity of Algorithm 1 is $\leq nm$.*

*Remark* 2.2 (State-of-the-art excess risk in linear time, without the restrictive assumptions). Under the assumptions that $\beta < (L\varepsilon^3/R)\sqrt{n^3/md^3}$ and $m \leq d/\varepsilon^2 \leq n$, [BS23] gave a linear-time algorithm with similar excess risk to Algorithm 1. However, their assumptions are very restrictive in practice: For example, in the canonical regime $n \approx d$, their assumption on $\beta$ rules out essentially every (non-linear) loss function. By contrast, our result holds even if the smoothness parameter is huge ($\beta \approx \sqrt{nmd}$) and we only require a logarithmic number of users. Thus, our algorithm and result is applicable to many practical ML problems.

To prove that Algorithm 1 is private, we essentially argue that for any phase $i$, the $\ell_2$-sensitivity of $\tilde{x}_i$ is upper bounded by $\widetilde{O}(\tau_i/C)$ with probability at least $1 - \delta/2$. The argument goes roughly as follows: First, the Laplace noise added to $s_i(\tau_i)$ ensures that $s_i(\tau_i)$ is $\varepsilon/4$-user-level DP. Now, it suffices to assume $\widehat{s}_i(\tau_i) \geq 4C/5$, since otherwise the algorithm halts and outputs 0 independently of the data. Next, conditional on the high probability event that the Laplace noise is smaller than $\widetilde{O}(1/\varepsilon)$, we know that $\widehat{s}_i(\tau_i) \geq 4C/5 \implies s_i(\tau_i) \geq 2C/3$ with high probability by our choice of $C$. In this case, an argument along the lines of [AL24, Lemma 3.5] shows that $\tilde{x}_i$ has sensitivity bounded by $\widetilde{O}(\tau_i/C)$ with probability at least $1 - \delta/2$. See Appendix B for the detailed proof.

To prove the excess risk bound in Algorithm 1, the key step is to show that if the data is i.i.d., then with high probability, no points are removed from $\{\tilde{x}_{i,j}\}_{j=1}^{C}$ during the outlier-removal phase

**Algorithm 1:** User-Level DP Phased SGD with Outlier Iterate Removal and Output Perturbation

1 **Input:** Dataset $\mathcal{D} = (Z_1, \ldots, Z_n)$, privacy parameters $(\varepsilon, \delta)$, parameters $p, q > 0$, stepsize $\eta$;
2 Set $l = \lfloor \log_2(n) \rfloor$, $C := 100 \log(20nme^\varepsilon/\delta)/\varepsilon$;
3 **for** $i = 1, \cdots, l$ **do**
4 $\quad$ Set $n_i = (1 - (1/2)^q)n/2^{iq}$, $\eta_i = \eta/2^{pi}$, $N_i = n_i/C$, $T_i = N_i m$,
$\quad\quad \tau_i = 1000\eta_i L\sqrt{T_i} \log(ndm)$;
5 $\quad$ Draw disjoint users $D_i$ of size $n_i$ from $\mathcal{D}$;
6 $\quad$ Divide $D_i$ into $C$ disjoint subsets $\{D_{i,j}\}_{j=1}^C$, each containing $|D_{i,j}| = N_i$ users;
7 $\quad$ **for** $j = 1, \cdots, C$ **do**
8 $\quad\quad$ $\tilde{x}_{i,j} \leftarrow SGD(D_{i,j}, \eta_i, T_i, x_{i-1}) = $ average iterate of $T_i$ steps of one-pass projected SGD
$\quad\quad\quad$ with data $D_{i,j}$, stepsize $\eta_i$, and initial point $x_{i-1}$ ;
9 $\quad$ **end**
10 $\quad$ Compute the concentration score for $D_i$:

$$s_i(\tau_i) := \frac{1}{C} \sum_{j,j' \in [C]} \mathbf{1}(\|\tilde{x}_{i,j} - \tilde{x}_{i,j'}\| \le \tau_i)$$

$\quad\quad$ Let $\hat{s}_i(\tau_i) = s_i(\tau_i) + \mathrm{Lap}(20/\varepsilon)$;
11 $\quad$ **if** $\hat{s}_i(\tau_i) \ge \frac{4C}{5}$ **then**
12 $\quad\quad$ $\mathcal{S}_i = \emptyset$ ;
13 $\quad\quad$ **for** $j = 1, \cdots, C$ **do**
14 $\quad\quad\quad$ Compute the score function of $\tilde{x}_{i,j}$: $h_{i,j} = \sum_{j'=1}^C \mathbf{1}(\|\tilde{x}_{i,j} - \tilde{x}_{i,j'}\| \le 2\tau_i)$;
15 $\quad\quad\quad$ Add $\tilde{x}_{i,j}$ to $\mathcal{S}_i$ with probability $p_{i,j}$ for $p_{i,j} = \begin{cases} 0 & h_{i,j} < C/2 \\ 1 & h_{i,j} \ge 2C/3 \\ \frac{h_{i,j} - C/2}{C/6} & o.w. \end{cases}$
16 $\quad\quad$ **end**
17 $\quad$ **end**
18 $\quad$ **else**
19 $\quad\quad$ **Halt; Output 0**
20 $\quad$ **end**
21 $\quad$ Let $\tilde{x}_i = \frac{1}{|\mathcal{S}_i|} \sum_{\tilde{x}_{i,j} \in \mathcal{S}_i} \tilde{x}_{i,j}$ ;
22 $\quad$ $x_i \leftarrow \tilde{x}_i + \zeta_i$, where $\zeta_i \sim \mathcal{N}(0, \sigma_i^2 I_d)$ with $\sigma_i = \frac{100\tau_i \log^2(n/\delta)}{\varepsilon C}$;
23 **end**
24 **Output:** $x_l$.

---

(i.e. $|\mathcal{S}_i| = C$). If $|\mathcal{S}_i| = C$ holds, then we can use the convergence guarantees of SGD and the localization arguments in [FKT20] to establish the excess risk guarantee. In order to prove that $|\mathcal{S}_i| = C$ with high probability, we need the following novel *stability* lemma:

**Lemma 2.3.** *Assume $f(\cdot, z)$ is convex, $L$-Lipschitz, and $\beta$-smooth on $\mathcal{X}$ with $\eta \le 1/\beta$. Let $\tilde{x} \leftarrow SGD(D, \eta, T, x_0)$ and $\tilde{y} \leftarrow SGD(D', \eta, T, x_0)$ be two independent runs of projected SGD, where $D, D' \sim P^N$ are i.i.d. Then, with probability at least $1 - \zeta$, we have*

$$\|\tilde{x} - \tilde{y}\| \lesssim \eta L \sqrt{T \log(dT/\zeta)}.$$

We prove Lemma 2.3 via induction on $t$, using non-expansiveness of gradient descent on smooth losses [HRS16], subgaussian concentration bounds, and a union bound.

Lemma 2.3 implies that if the data is i.i.d., then the following events hold with high probability: $\|\tilde{x}_{i,j} - \tilde{x}_{i,j'}\| \le \tau_i$ for all $j, j' \in [C_i]$ and hence $s_i(\tau_i) = C$. Further, conditional on $s_i(\tau_i) = C$, we know that $\hat{s}_i(\tau_i) \ge 4C/5$ with high probability, so that the algorithm does not halt. Also, $\|\tilde{x}_{i,j} - \tilde{x}_{i,j'}\| \le \tau_i$ for all $j, j'$ implies $p_{i,j} = 1$ for all $j$ and hence $|\mathcal{S}_i| = C$ for all $j$. The detailed excess risk proof is provided in Appendix B.

**Challenges of getting optimal excess risk in linear time:** In the item-level DP setting, there are several (nearly) linear time algorithms that achieve optimal excess risk for smooth DP SCO

under mild smoothness conditions, such as snowball SGD [FKT20], phased SGD [FKT20], and phased ERM with output perturbation [ZTOH22]. Extending these approaches into optimal nearly linear-time user-level DP algorithms is challenging. First, the user-level DP implementation of output perturbation in [GKK+23] is computationally inefficient. Second, snowball SGD relies on *privacy amplification by iteration*, which does not extend nicely to the user-level DP case due to instability of the outlier-detection procedure in [AL24]. Specifically, since amplification by iteration intermediate only provides DP for the last iterate $x_T$ but not the intermediate iterates $x_t$ ($t < T$), the sensitivity of the concentration score function is not $O(1)$, which impairs DP outlier-detection. A similar instability issue arises if one tries to naively extend phased SGD to be user-level DP by applying [AL24] to user gradients. This issue motivates our Algorithm 1, which extends phased SGD in an alternative way: by applying outlier-detection/removal to the SGD *iterates* instead of the gradients, we can control the sensitivity of the concentration score and thus prove that our algorithm is DP. However, since the bound in Lemma 2.3 scales polynomially with $T$ (and we believe this dependence on $T$ is necessary), Algorithm 1 adds excessive noise and has suboptimal excess risk. We believe that obtaining optimal risk in linear time will require a fundamentally different user-level DP mean estimation procedure that does not suffer from the instability issue.

# 3 An optimal algorithm with $\approx (mn)^{9/8}$ gradient complexity for smooth losses

In this section, we provide an algorithm that achieves optimal excess risk using $\approx (mn)^{9/8}$ stochastic gradient evaluations. Our Algorithm 3 is inspired by the item-level accelerated phased ERM algorithm of [KLL21]. It applies iterative localization [FKT20] to the user-level DP accelerated minibatch SGD Algorithm 2. Algorithm 2 is a user-level DP variation of the accelerated minibatch SGD of [GL12, GL13].

Our Algorithm 2 applies a DP outlier-removal procedure to the users' gradients in each iteration. We use *Above Threshold* [DR14] to privatize the concentration scores $s_i^{(t)}$ and determine whether or not most of the gradients of users in minibatch $D_i^t$ are $2\tau$-close to each other. If $\widehat{s}_i^t \geq \widehat{\Delta}_i$, indicating that the gradients of users in $D_i^t$ are nearly $2\tau$-concentrated, then we proceed with outlier removal in lines 8-12. We then invoke *privacy amplification by subsampling* [BBG18] and the *advanced composition theorem* [KOV15] to privatize the average of the "inlier" gradients with additive Gaussian noise. By properly choosing algorithmic parameters, we obtain the following results, proved in Appendix C:

**Theorem 3.1** (Privacy of Algorithm 3)**.** *Let $\varepsilon \leq 10$, $q > 0$ such that $n^{1-q} > \frac{100 \log(20nmde^\varepsilon/\delta)}{\varepsilon(1-(1/2)^q)}$. Then, Algorithm 3 is $(\varepsilon, \delta)$-DP.*

**Theorem 3.2** (Utility & runtime of Algorithm 3 - Informal)**.** *Let $\varepsilon \leq 10$ and $\delta < 1/(mn)$. Then, Algorithm 3 yields optimal excess risk:*

$$\mathbb{E}F(x_l) - F^* \leq LR \cdot \widetilde{O}\left(\frac{1}{\sqrt{mn}} + \frac{\sqrt{d\log(1/\delta)}}{\varepsilon n \sqrt{m}}\right).$$

*The gradient complexity of this algorithm is upper bounded by*

$$mn\left(1 + \varepsilon\left(\frac{\beta R}{L}\right)^{1/4}\left((mn)^{1/8} \wedge \left(\frac{\varepsilon^2 n^2 m}{d}\right)^{1/8}\right)\right) + \sqrt{\frac{\beta R}{L}}\left(\frac{n^{1/4}m^{5/4}}{\varepsilon} + \left(\frac{n^{1/2}m^{5/4}}{d^{1/4}\varepsilon^{1/2}}\right)\right).$$

*If $n = d$, $\varepsilon = 1$, and $\beta R = L$ then the gradient complexity bound in Theorem 3.2 simplifies to $(mn)^{9/8} + n^{1/4}m^{5/4}$. Typically, $n^7 \geq m$, so that the dominant term in this bound is $(mn)^{9/8}$.*

*Remark* 3.3 (State-of-the-art runtime)*.* The gradient complexity bound in Theorem 3.2 is *superior to the runtime bounds of all existing near-optimal algorithms by polynomial factors* in $n, m$, and $d$ [BS23, GKK+23, AL24]. Note that while the $mn^{3/2}$ gradient complexity bound of [BS23] may *appear* to be better than $\beta^{1/4}(nm)^{9/8}$ in certain parameter regimes (e.g. $m > n^3$ or $\beta \gg nm$), this is not the case: the result of [BS23] requires $m < n$ and $\beta < \sqrt{n/m}$.

*Remark* 3.4 (Mild assumptions)*.* Note that Theorems 3.1 and 3.2 do not require any bound on the smoothness parameter $\beta$, and only require the number of users to grow logarithmically: $n^{1-o(1)} \geq \widetilde{\Omega}(1/\varepsilon)$. Contrast this with the results of previous works (e.g. [BS23]).

**Algorithm 2:** User-Level DP Accelerated Minibatch SGD$(\widehat{F}_i, T_i, K_i, x_{i-1}, \tau, \varepsilon, \delta)$

**1** Initialize $x_{i-1}^1 \leftarrow x_{i-1}$;

**2 for** $t = 1, \cdots, T_i$ **do**

**3** $\quad$ Draw $K_i$ random users $D_i^t = \{Z_{i,j}^t\}_{j=1}^{K_i}$ from $D_i$ uniformly with replacement;

**4** $\quad$ Set noisy threshold $\widehat{\Delta}_i := \frac{4K_i}{5} + \xi_i$, where $\xi_i \sim \mathrm{Lap}\left(\frac{8}{\varepsilon}\right)$;

**5** $\quad$ Let $q_t(Z) := \frac{1}{m} \sum_{z \in Z} \nabla f(x_{i-1}^t, z)$ for user $Z$;

**6** $\quad$ Compute the concentration score of $D_i^t$:

$$s_i^t(\tau) := \frac{1}{K_i} \sum_{Z, Z' \in D_i^t} \mathbf{1}(\|q_t(Z) - q_t(Z')\| \le 2\tau)$$

$\quad$ Let $\widehat{s}_i^t(\tau) = s_i^t(\tau_i) + v_i^t$, where $v_i^t \sim \mathrm{Lap}\left(\frac{16}{\varepsilon}\right)$;

**7** $\quad$ **if** $\widehat{s}_i^t(\tau) \ge \widehat{\Delta}_i$ **then**

**8** $\quad\quad$ $\mathcal{S}_i^t = \emptyset$;

**9** $\quad\quad$ **for** *Each User* $Z \in D_i^t$ **do**

**10** $\quad\quad\quad$ Set $h_i^t(Z) = \sum_{Z' \in D_i^t} \mathbf{1}(\|q_t(Z) - q_t(Z')\| \le 2\tau)$;

**11** $\quad\quad\quad$ Add $Z$ to $\mathcal{S}_i^t$ with probability $p_i^t(Z) := \begin{cases} 0 & h_i^t(Z) < K_i/2 \\ 1 & h_i^t(Z) \ge 2K_i/3 \\ \frac{h_i^t(Z) - K_i/2}{K_i/6} & o.w. \end{cases}$

**12** $\quad\quad$ **end**

**13** $\quad\quad$ $g_i^t = \frac{1}{|\mathcal{S}_i^t|} \sum_{Z \in \mathcal{S}_i^t} \nabla \widehat{F}(x_{i-1}^t, Z)$;

**14** $\quad\quad$ $\widehat{g}_i^t = g_i^t + \zeta_i^t$, where $\zeta_i^t \sim \mathcal{N}(0, \sigma_i^2)$ with $\sigma_i = \frac{1000\tau \sqrt{T_i} \log(nde^\varepsilon/\delta)}{\varepsilon n_i}$ [GL12];

**15** $\quad\quad$ Do 1 iteration of Accelerated Minibatch SGD (AC-SA) [GL12] on $\widehat{F}_i$, using gradient estimator $\widehat{g}_i^t + \lambda_i(x_{i-1}^t - x_{i-1})$ to obtain $x_{i-1}^{t+1}$.

**16** $\quad$ **end**

**17** $\quad$ **else**

**18** $\quad\quad$ **Halt; Return 0**

**19** $\quad$ **end**

**20 end**

**21 Output** $x_{i-1}^{T_i}$.

---

**Algorithm 3:** User-Level DP Accelerated Phased ERM with Outlier Gradient Removal

**1 Input:** Dataset $\mathcal{D} = (Z_1, \ldots, Z_n)$, privacy parameters $(\varepsilon, \delta)$, parameters $p, q, \lambda > 0$;

**2** Set $l = \lfloor \log_2(n) \rfloor$ and $\tau = O(L \log(ndm)/\sqrt{m})$, choose any initial point $x_0 \in \mathcal{X}$;

**3 for** $i = 1, \cdots, l$ **do**

**4** $\quad$ Set $n_i = (1 - (1/2)^q)n/2^{iq}$, $\lambda_i = \lambda \cdot 2^{pi}$, $T_i = \widetilde{\Theta}(1 + \sqrt{\beta/\lambda_i})$,

$\quad\quad K_i = 500 \log(n_i^2 m^2 e^\varepsilon/\delta) \left(\frac{1}{\varepsilon} + \frac{n_i \varepsilon}{\sqrt{T_i \log(1/\delta)}}\right)$;

**5** $\quad$ Draw disjoint users $D_i$ of size $n_i$ from $\mathcal{D}$;

**6** $\quad$ Let $\widehat{F}_i(x) := \frac{1}{n_i} \sum_{Z_{i,j} \in D_i} \widehat{F}(x, Z_{i,j}) + \frac{\lambda_i}{2}\|x - x_{i-1}\|^2$, where $\widehat{F}(x, Z_{i,j})$ is user $Z_{i,j}$'s empirical loss;

**7** $\quad$ $x_i \leftarrow$ User-Level DP Accelerated Minibatch SGD$(\widehat{F}_i, T_i, K_i, x_{i-1}, \tau, \varepsilon, \delta)$. ;

**8 end**

**9 Output** $x_l$.

A challenge in proving Theorem 3.2 is getting a tight bound on the variance of the the noisy minibatch stochastic gradients $\widehat{g}_i^t$ that are used in Algorithm 2 (lines 12-14). Conditional on $\mathcal{S}_i^t = D_i^t$, it is easy to obtain a variance bound of the form $\mathbb{E}\|\widehat{g}_i^t - \nabla\widehat{F}_i(x_i^t)\|^2 \lesssim d\sigma_i^2 + \frac{L^2}{K_i}$, since we are sampling $K_i$ users uniformly at random. However, this bound is too weak to obtain Theorem 3.2, since it does not scale with $m$. To prove Theorem 3.2, we need the following stronger result:

**Lemma 3.5** (Variance Bound for Algorithm 2)**.** *Let* $\delta \leq 1/(nm), \varepsilon \lesssim 1$. *Denote* $\widetilde{F}_i(x) :=$ $\frac{1}{n_i}\sum_{Z_{i,j}\in D_i}\widehat{F}(x, Z_{i,j})$. *Then, conditional on* $\mathcal{S}_i^t = D_i^t$ *for all* $i \in [l], t \in [T_i]$, *we have*

$$\mathbb{E}\|g_i^t - \nabla\widetilde{F}_i(x_{i-1}^t)\|^2 \lesssim \frac{L^2\log(ndm)}{Km}$$

*for all* $i \in [l], t \in [T_i]$, *where the expectation is over both the random i.i.d. draw of* $\mathcal{D} = (Z_1, \ldots, Z_n) \sim P^{nm}$ *and the randomness in Algorithm 3.*

The difficulty in proving Lemma 3.5 comes from the fact that the iterates $x_i^t$ and the data $\mathcal{D}$ are not independent. To overcome this difficulty, we use the *stability of user-level DP* [BS23] to argue that for all $Z \in D_i$, $\nabla\widehat{F}(x_{i-1}^t, Z)$ is $\approx L/\sqrt{m}$-close to $\nabla F(x_{i-1}^t)$ with high probability, since $x_{i-1}^t$ is user-level DP. A detailed proof is given in Appendix C.

*Remark* 3.6 (Strongly convex losses: Optimal excess risk with state-of-the-art runtime). If $f(\cdot, z)$ is $\mu$-strongly convex, then Algorithm 3 can be combined with the meta-algorithm of [FKT20, Section 5.1] to obtain optimal excess risk

$$\frac{L^2}{\mu} \cdot \widetilde{O}\left(\frac{1}{nm} + \frac{d\ln(1/\delta)}{\varepsilon^2 n^2 m}\right)$$

with the same gradient complexity stated in Theorem 3.2. This improves over the previous state-of-the-art gradient complexity $\approx \beta(mn)^{3/2}$ of [AL24].

## 4 An optimal algorithm with subquadratic gradient complexity for non-smooth losses

In this section, we extend our accelerated algorithm from the previous section to non-smooth loss functions. To accomplish this with minimal computational cost, we apply *randomized (convolution) smoothing* [YNS12, DBW12] to approximate non-smooth $f$ by a $\beta$-smooth $\tilde{f}$. We can then apply Algorithm 3 to $\tilde{f}$. Since convolution smoothing is by now a standard optimization technique, we defer the details and proof to Appendix D.

**Theorem 4.1** (Privacy and utility of smoothed Algorithm 3 for non-smooth loss - informal)**.** *Let* $\varepsilon \leq 10$, $\delta < 1/(mn)$, *and* $q > 0$ *such that* $n^{1-q} > \frac{100\log(20nmde^\varepsilon/\delta)}{\varepsilon(1-(1/2)^q)}$. *Then, applying Algorithm 3 to the smooth approximation of* $f$ *yields optimal excess risk:*

$$\mathbb{E}F(x_l) - F^* \leq LR \cdot \widetilde{O}\left(\frac{1}{\sqrt{mn}} + \frac{\sqrt{d\log(1/\delta)}}{\varepsilon n\sqrt{m}}\right).$$

*The gradient complexity of this algorithm is upper bounded by*

$$mn\left(1 + n^{3/8}m^{1/4}\varepsilon^{1/4}\right).$$

*Remark* 4.2 (State-of-the-art gradient complexity). The only previous polynomial-time algorithm that can achieve optimal excess risk for non-smooth loss functions is due to [AL24]. The algorithm of [AL24] required $(nm)^3 + (mn)^2\sqrt{d}$ gradient evaluations. Thus, the gradient complexity of the smoothed version of Algorithm 3 offers a *significant improvement over the previous state-of-the-art*. For example, if $\varepsilon = 1$, then our algorithm is faster than the previous state-of-the-art by a multiplicative factor of at least $n^{13/8}m^{7/4}$.

## 5 Conclusion

In this paper, we developed new user-level DP algorithms with improved runtime and excess risk guarantees for stochastic convex optimization without the restrictive assumptions made in prior

works. Our accelerated Algorithm 3 achieves optimal excess risk for both smooth and non-smooth loss functions, with significantly smaller computational cost than the previous state-of-the-art. Our linear-time Algorithm 1 achieves state-of-the-art excess risk under much milder, more practical assumptions than existing linear-time approaches.

Our work paves the way for several intriguing future research directions. First, the question of whether there exists a linear-time algorithm that can attain the user-level DP lower bound for smooth losses remains open. In light of our improved gradient complexity bound ($\approx (nm)^{9/8}$), we are now optimistic that the answer to this question is "yes." We believe that our novel techniques will be key to the development of an optimal linear-time algorithm. Specifically, utilizing Lemma 2.3 to apply outlier removal to the iterates instead of the gradients appears to be pivotal. Second, the study of user-level DP SCO has been largely limited to approximate $(\varepsilon, \delta)$-DP. What rates are achievable under the stronger notion of pure $\varepsilon$-user-level DP? Third, it would be useful to develop fast and optimal algorithms that are tailored to federated learning environments [MRTZ18, GLZW24], where only a small number of users may be available to communicate with the server in each iteration. We hope our work inspires and guides further research in this exciting and practically important area.

## Acknowledgements

AL's research is supported by NSF grant 2023239 and the AFOSR award FA9550-21-1-0084. We thank the anonymous NeurIPS reviewers for their helpful feedback.

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

# Appendix

## A   More Preliminaries

### A.1   Tools from Differential Privacy

**Additive Noise Mechanisms**   Additive noise mechanisms privatize a query by adding noise to its output, with the scale of the noise calibrated to the *sensitivity* of the query.

**Definition A.1** (Sensitivity). Given a function $q : \mathcal{Z}^N \to \mathbb{R}^k$ and a norm $\|\cdot\|_p$ on $\mathbb{R}^k$, the $\ell_p$-*sensitivity* of $q$ is defined as

$$\sup_{\mathcal{D} \sim \mathcal{D}'} \|q(\mathcal{D}) - q(\mathcal{D}')\|_p,$$

where the supremum is taken over all pairs of datasets that differ in one user's data.

**Definition A.2** (Laplace Distribution). We say $X \sim \text{Lap}(b)$ if the density of $X$ is $f(X = x) = \frac{1}{2b}\exp(-\frac{|x|}{b})$.

**Definition A.3** (Laplace Mechanism). Let $\varepsilon > 0$. Given a function $q : \mathcal{Z}^N \to \mathbb{R}^k$ on $\mathbb{R}^k$ with $\ell_1$-sensitivity $\Delta$, the *Laplace Mechanism* $\mathcal{M}$ is defined by

$$\mathcal{M}(\mathcal{D}) := q(\mathcal{D}) + (Y_1, \dots, Y_k),$$

where $\{Y_i\}_{i=1}^k$ are i.i.d., $Y_i \sim \text{Lap}\left(\frac{\Delta}{\varepsilon}\right)$.

**Lemma A.4** (Privacy of Laplace Mechanism [DR14]). *The Laplace Mechanism is $\varepsilon$-DP.*

**Definition A.5** (Gaussian Mechanism). Let $\varepsilon > 0, \delta \in (0, 1)$. Given a function $q : \mathcal{Z}^N \to \mathbb{R}^k$ with $\ell_2$-sensitivity $\Delta$, the *Gaussian Mechanism* $\mathcal{M}$ is defined by

$$\mathcal{M}(\mathcal{D}) := q(\mathcal{D}) + G$$

where $G \sim \mathcal{N}_k\left(0, \sigma^2 \mathbf{I}_k\right)$ and $\sigma^2 = \frac{2\Delta^2 \log(2/\delta)}{\varepsilon^2}$.

**Lemma A.6** (Privavcy of Gaussian Mechanism [DR14]). *The Laplace Mechanism is $(\varepsilon, \delta)$-DP.*

**Advanced Composition**   If we adaptively query a data set $T$ times, then the privacy guarantees of the $T$-th query is still DP and the privacy parameters degrade gracefully:

**Lemma A.7** (Advanced Composition Theorem [DR14]). *Let $\varepsilon \geq 0, \delta, \delta' \in [0, 1)$. Assume $\mathcal{A}_1, \cdots, \mathcal{A}_T$, with $\mathcal{A}_t : \mathcal{Z}^n \times \mathcal{X} \to \mathcal{X}$, are each $(\varepsilon, \delta)$-DP $\forall t = 1, \cdots, T$. Then, the adaptive composition $\mathcal{A}(\mathcal{D}) := \mathcal{A}_T(\mathcal{D}, \mathcal{A}_{T-1}(\mathcal{D}, \mathcal{A}_{T-2}(X, \cdots)))$ is $(\varepsilon', T\delta + \delta')$-DP for*

$$\varepsilon' = \sqrt{2T \ln(1/\delta')}\varepsilon + T\varepsilon(e^\varepsilon - 1).$$

**Privacy Amplification by Subsampling**

**Lemma A.8** ([Ull17]). *Let $\mathcal{M} : \mathcal{Z}^M \to \mathcal{X}$ be $(\varepsilon, \delta)$-DP. Let $\mathcal{M}' : \mathcal{Z}^N \to \mathcal{X}$ that first selects a random subsample $\mathcal{D}'$ of size $M$ from the data set $\mathcal{D} \in \mathcal{Z}^N$ and then outputs $\mathcal{M}(\mathcal{D}')$. Then, $\mathcal{M}'$ is $(\varepsilon', \delta')$-DP, where $\varepsilon' = \frac{(e^\varepsilon - 1)M}{N}$ and $\delta' = \frac{\delta M}{N}$.*

**AboveThreshold:**   AboveThreshold algorithm [DR14] which is a key tool in differential privacy to identify whether there is a query $q_i : \mathcal{Z} \to \mathbb{R}$ in a stream of queries $q_1, \dots, q_T$ that is above a certain threshold $\Delta$. The AboveThreshold Algorithm 4 has the following guarantees:

**Lemma A.9** ([DR14], Theorem 3.24). *Let $\gamma > 0$ and $\alpha = \frac{8 \log(2T/\gamma)}{\varepsilon}$, $k \in [T+1]$. AboveThreshold is $(\varepsilon, 0)$-DP. Moreover, with probability at least $1 - \gamma$, for all $t \leq k$, we have:*

- *if $a_t = \top$, then $q_t(\mathcal{D}) \geq \Delta - \alpha$; and*

- *if $a_t = \bot$, then $q_t(\mathcal{D}) \leq \Delta + \alpha$.*

---
**Algorithm 4:** AboveThreshold
---
1 **Input:** Dataset $\mathcal{D} = (Z_1, \ldots, Z_n)$, threshold $\Delta \in \mathbb{R}$, privacy parameter $\varepsilon$, sequence of $T$ queries $q_1, \cdots, q_T : \mathcal{Z}^n \to \mathbb{R}$, each with $\ell_1$-sensitivity 1;

2 Let $\widehat{\Delta} := \Delta + \mathrm{Lap}(\frac{2}{\varepsilon})$;

3 **for** $t = 1$ *to* $T$ **do**

4      Receive a new query $q_t : \mathcal{Z}^n \to \mathbb{R}$ ;

5      Sample $\nu_i \sim \mathrm{Lap}(\frac{4}{\varepsilon})$;

6      **if** $q_t(\mathcal{D}) + \nu_i \geq \widehat{\Delta}$ **then**

7          **Output:** $a_t = \top$;

8          **Halt**;

9      **end**

10      **else**

11          **Output:** $a_t = \bot$;

12      **end**

13 **end**
---

## A.2    SubGaussian and Norm-SubGuassian Random Vectors

**Definition A.10.** Let $\zeta > 0$. We say a random vector $X$ is *SubGaussian* (SG($\zeta$)) with parameter $\zeta$ if $\mathbb{E}[e^{\langle v, X - \mathbb{E}X \rangle}] \leq e^{\|v\|^2 \zeta^2 / 2}$ for any $v \in \mathbb{R}^d$. Random vector $X \in \mathbb{R}^d$ is *Norm-SubGaussian* with parameter $\zeta$ (nSG($\zeta$)) if $\mathbb{P}[\|X - \mathbb{E}X\| \geq t] \leq 2e^{-\frac{t^2}{2\zeta^2}}$ for all $t > 0$.

**Theorem A.11** (Hoeffding-type inequality for norm-subGaussian, [JNG$^+$19]). *Let $X_1, \cdots, X_k \in \mathbb{R}^d$ be random vectors, and let $\mathcal{F}_i = \sigma(x_1, \cdot, x_i)$ for $i \in [k]$ be the corresponding filtration. Suppose for each $i \in [k]$, $X_i \mid \mathcal{F}_{i-1}$ is zero-mean nSG($\zeta_i$). Then, there exists an absolute constant $c > 0$, for any $\gamma > 0$,*

$$\mathbb{P}\left[\left\|\sum_{i \in [k]} X_i\right\| \geq c\sqrt{\log(d/\gamma) \sum_{i \in [k]} \zeta_i^2}\right] \leq \gamma.$$

**Lemma A.12** ([JNG$^+$19]). *There exists an absolute constant $c$, such that if $X$ is nSG($\zeta$), then for any fixed unit vector $v \in \mathbb{R}^d$, $\langle v, X \rangle$ is $c\zeta$ norm-SubGaussian.*

# B    Proof of Theorem 2.1

**Theorem B.1** (Formal statement of Theorem 2.1). *Suppose $n^{1-q} \geq (100/(1-1/2^q))\log(n/\delta)/\varepsilon$ for some small $q > 0$, and $m \leq n^J$ for some large $J > 0$. Choose $p = J + 3/2$ and $\eta = R/(L\sqrt{dmn}\varepsilon)$. in Algorithm 1. Then, Algorithm 1 is $(\varepsilon, \delta)$-user-level DP and achieves excess risk*

$$\mathbb{E}F(x_l) - F^* \leq LR \cdot \widetilde{O}\left(\frac{1}{\sqrt{nm}\varepsilon} + \frac{\sqrt{d\log(1/\delta)}}{\sqrt{n}\varepsilon\sqrt{m}}\right),$$

*using $nm$ gradient evaluations, provided $\beta \leq (L/R)\sqrt{dmn}\varepsilon$.*

The gradient complexity is clear by inspection of the algorithm: The number of stochastic gradients computed during the algorithm is

$$\sum_{i=1}^{l} T_i C = \sum_{i=1}^{l} N_i m C = \sum_{i=1}^{l} n_i m \leq nm.$$

Next, we will prove the privacy statement in Theorem B.1. The following lemma ensures that if the Laplace noise added in Algorithm 1 is sufficiently small and outlier detection succeeds, then the sensitivity of $\tilde{x}_i$ is $\widetilde{O}(\tau_i/C)$ with high probability.

**Lemma B.2.** *[AL24, Slight modification of Lemma 3.5] Let $i \in [l]$ and $\zeta > 0$. Suppose $\mathcal{D}_i$ and $\mathcal{D}_i'$ differ in the data of one user and we are in phase $i$ of Algorithm 1. Let $E_i$ be the event that the Laplace noise added to the concentration score $s_i(\tau_i)$ for $\mathcal{D}_i$ has absolute value less than $2C/15$ and define $E_i'$ similarly for data $\mathcal{D}_i'$. Denote $a_i := \mathbf{1}(\widehat{s}_i(\tau_i) \geq 4C/5)$ and $a_i' := \mathbf{1}(\widehat{s}_i'(\tau_i) \geq 4C/5)$, where $\widehat{s}_i(\tau_i)$ and $\widehat{s}_i'(\tau_i)$ are the noisy concentration scores that we get when running phase $i$ of Algorithm 1 on neighboring $\mathcal{D}_i$ and $\mathcal{D}_i'$, respectively. Then, conditional on $a_i = a_i'$ and $E_i \bigcap E_i'$, there is a coupling $\Gamma_i$ over $\tilde{x}_i$ and $\tilde{x}_i'$ such that for $(y_i, y_i')$ drawn from $\Gamma_i$, we have*

$$\|y_i - y_i'\| \lesssim \frac{\tau_i \log(1/\zeta)}{C}$$

*with probability at least $1 - \zeta$.*

With Lemma B.2 in hand, we proceed to prove that Algorithm 1 is $(\varepsilon, \delta)$-user-level DP:

*Proof of Theorem B.1 - Privacy.* **Privacy:** Since the $\{D_i\}_{i=1}^l$ are disjoint, parallel composition of DP [McS09] implies that it suffices to prove that phase $i$ is $(\varepsilon, \delta)$-user-level-DP for any fixed $i$ and fixed $x_{i-1}$. To that end, let $\mathcal{D}$ and $\mathcal{D}'$ be adjacent datasets differing in the data of one user, say $Z_{i,1} \neq Z_{i,1}'$ without loss of generality. We will show that the outputs of phase $i$ when run on $\mathcal{D}$ and $\mathcal{D}'$, $x_i := x_i(\mathcal{D})$ and $x_i' := x_i(\mathcal{D}')$ respectively, are $(\varepsilon, \delta)$-indistinguishable.

Let $E_i$ be the event that the Laplace noise added in phase $i$ (for data set $\mathcal{D}$) has absolute value less than $2C/15$ and define $E_i'$ analogously for data set $\mathcal{D}'$. Note that $E_i$ and $E_i'$ are independent and $\mathbb{P}(E_i, E_i') \geq 1 - \delta/10e^\varepsilon$. Denote $\zeta := \delta/10e^\varepsilon$. Let $a_i := \mathbf{1}(\widehat{s}_i(\tau_i) \geq 4C/5)$ and $a_i' := \mathbf{1}(\widehat{s}_i'(\tau_i) \geq 4C/5)$, where $\widehat{s}_i(\tau_i)$ and $\widehat{s}_i'(\tau_i)$ are the noisy concentration scores that we get when running phase $i$ of Algorithm 1 on neighboring $\mathcal{D}_i$ and $\mathcal{D}_i'$, respectively. By Lemma B.2 and our choice of $C$, we know that, conditional on $E_i \bigcap E_i'$ and on $a_i = a_i'$, there exists a coupling $\Gamma$ over $(\tilde{x}_i, \tilde{x}_i')$ such that for $(y_i, y_i')$ drawn from $\Gamma$, we have

$$\|y_i - y_i'\| \lesssim \frac{\tau_i \log(1/\zeta)}{C} \tag{1}$$

with probability at least $1 - \zeta$.

Note that the sensitivity of $s_i$ is less than or equal to 2. Thus, by the privacy guarantees of the Laplace mechanism (Lemma A.4), we have

$$\mathbb{P}(a_i = b) \leq e^{\varepsilon/4}\mathbb{P}(a_i' = b) \tag{2}$$

for any $b \in \{0, 1\}$. Further, this implies

$$\mathbb{P}(a_i = b, E_i) \leq e^{\varepsilon/4}\left[\mathbb{P}(a_i' = b, E_i') + \zeta\right]. \tag{3}$$

By the bound (1), the privacy guarantee of the Gaussian mechanism (Lemma A.6), our choice of $\sigma_i$, and independence of the Laplace and Gaussian noises that we add in Algorithm 1, we have

$$\mathbb{P}(x_i \in \mathcal{O} \mid E_i, a_i = 1) \leq e^{\varepsilon/4}\mathbb{P}(x_i' \in \mathcal{O} \mid E_i', a_i' = 1) + \frac{\delta}{n} + \zeta, \tag{4}$$

for any event $\mathcal{O} \subset \mathcal{X}$.

Moreover, since the algorithm halts and returns $x_i = 0$ if $a_i = 0$, we know that

$$\mathbb{P}(x_i \in \mathcal{O} \mid E_i, a_i = 0) = \mathbb{P}(x_i' \in \mathcal{O} \mid E_i', a_i' = 0) \tag{5}$$

for any event $\mathcal{O} \subset \mathcal{X}$.

Therefore,

$$
\begin{aligned}
\mathbb{P}(x_i \in \mathcal{O}) &= \mathbb{P}(x_i \in \mathcal{O} \mid E_i)\mathbb{P}(E_i) + \mathbb{P}(x_i \in \mathcal{O} \mid E_i^c)\mathbb{P}(E_i^c) \\
&\leq \mathbb{P}(x_i \in \mathcal{O} \mid E_i, a_i = 1)\mathbb{P}(E_i, a_i = 1) + \mathbb{P}(x_i \in \mathcal{O} \mid E_i, a_i = 0)\mathbb{P}(E_i, a_i = 0) + \zeta \\
&\overset{(i)}{\leq} e^{\varepsilon/4}\mathbb{P}(x_i' \in \mathcal{O} \mid E_i', a_i' = 1)e^{\varepsilon/4}\left[\mathbb{P}(E_i', a_i' = 1) + \zeta\right] \\
&\quad + \mathbb{P}(x_i' \in \mathcal{O} \mid E_i', a_i' = 0)e^{\varepsilon/4}\left[\mathbb{P}(E_i', a_i' = 0) + \zeta\right] + \zeta
\end{aligned}
$$

$$\leq e^{\varepsilon/2}\mathbb{P}(x_i' \in \mathcal{O}, E_i') + \zeta\left(2e^{\varepsilon/2} + 1\right)$$
$$\leq e^{\varepsilon}\mathbb{P}(x_i' \in \mathcal{O}) + \delta,$$

where $(i)$ follows from inequalities (3), (4), and (5). Thus, $x_i$ is $(\varepsilon, \delta)$-user-level-DP. This completes the privacy proof.

$\square$

Next, we turn to the excess risk proof. The following lemma is immediate from [FKT20, Lemma 4.5]:

**Lemma B.3.** *Let $\eta_i \leq 1/\beta$. Then, for any $y \in \mathcal{X}$ and all $i, j$, we have*

$$\mathbb{E}[F(\tilde{x}_{i,j}) - F(y)] \leq \frac{\mathbb{E}\|y - x_{i-1}\|^2}{\eta_i T_i} + \eta_i L^2.$$

The next novel lemma is crucial in our analysis:

**Lemma B.4** (Re-statement of Lemma 2.3). *Assume $f(\cdot, z)$ is convex, $L$-Lipschitz, and $\beta$-smooth on $\mathcal{X}$ with $\eta \leq 1/\beta$. Let $\tilde{x} \leftarrow SGD(D, \eta, T, x_0)$ and $\tilde{y} \leftarrow SGD(D', \eta, T, x_0)$ be two independent runs of projected SGD, where $D, D' \sim P^N$ are i.i.d. Then, with probability at least $1 - \zeta$, we have*

$$\|\tilde{x} - \tilde{y}\| \lesssim \eta L \sqrt{T \log(dT/\zeta)}.$$

*Proof.* Let $g_t := \nabla f(x_t, z_t)$ for $z_t$ drawn uniformly from $D$ without replacement and $g_t' := \nabla f(y_t, z_t')$ for $z_t'$ drawn uniformly from $D'$ without replacement. Let $F(x) := \mathbb{E}_{z \sim P}[f(x, z)]$.

We will prove that $\|x_t - y_t\| \lesssim \eta L \sqrt{T \log(dT/\zeta)}$ with probability at least $1 - \zeta/t$ for all $t \in [T]$. Note that this implies the lemma. We proceed by induction. The base case, when $t = 0$, is trivially true since $x_0 = y_0$. For the inductive hypothesis, suppose there is an absolute constant $c > 0$ such that with probability at least $1 - t\zeta/T$, we have

$$\|x_i - y_i\| \leq c\eta L \sqrt{i \cdot \log(dT/\zeta)} + 2\eta L,$$

$\forall i \leq t$. Then, for the inductive step, we have by non-expansiveness of projection onto convex sets, that

$$\begin{aligned}
\|x_{t+1} - y_{t+1}\|^2 &\leq \|x_t - \eta g_t - (y_t - \eta g_t')\|^2 \\
&= \|x_t - \eta\nabla F(x_t) - (y_t - \eta\nabla F(y_t)) - \eta(g_t - \nabla F(x_t) - g_t' + \nabla F(y_t))\|^2 \\
&= \|x_t - \eta\nabla F(x_t) - (y_t - \eta\nabla F(y_t))\|^2 \\
&\quad - 2\eta\langle x_t - \eta\nabla F(x_t) - (y_t - \eta\nabla F(y_t)), g_t - \nabla F(x_t) - g_t' + \nabla F(y_t)\rangle \\
&\quad + \eta^2\|g_t - \nabla F(x_t) - g_t' + \nabla F(y_t)\|^2 \\
&\overset{(i)}{\leq} \|x_t - y_t\|^2 - 2\eta\langle x_t - \eta\nabla F(x_t) - (y_t - \eta\nabla F(y_t)), g_t - \nabla F(x_t) - g_t' + \nabla F(y_t)\rangle \\
&\quad + 4\eta^2 L^2,
\end{aligned}$$
$\qquad\qquad\qquad\qquad\qquad\qquad\qquad\qquad\qquad\qquad\qquad\qquad\qquad\qquad\qquad (6)$

where $(i)$ follows from the non-expansive property of gradient descent on smooth convex function for $\eta \leq 1/\beta$ [HRS16].

Define $a_t := -2\eta\langle x_t - \eta\nabla F(x_t) - (y_t - \eta\nabla F(y_t)), g_t - \nabla F(x_t) - g_t' + \nabla F(y_t)\rangle$. By Inequality (6) and the inductive hypothesis, we obtain

$$\|x_{t+1} - y_{t+1}\|^2 \leq 4\eta^2 L^2 t + \sum_{i=1}^{t} a_t.$$

It remains to bound $\sum_{i=1}^{t} a_i$. Note that $\mathbb{E}[a_i \mid a_1, \cdots, a_{i-1}] = 0$, and by Lemma A.12 we know there is a constant $c > 0$ such that $a_i$ is $\mathrm{nSG}(c\eta L\|x_i - y_i\|)$ for all $i$. Hence by Theorem A.11, we know

$$\mathbb{P}\left[\left|\sum_{i=1}^{t} a_i\right| \geq c\eta L \sqrt{\log(dT/\gamma) \sum_{i \leq t} \|x_i - y_i\|^2}\right] \leq 1 - \zeta/T.$$

Conditional on the event that $\|x_i - y_i\| \leq c\sqrt{\log(dT/\zeta)}\eta L\sqrt{i}$ for all $i \leq t$ (which happens with probability $1 - t\zeta/T$ by the inductive hypothesis), we know

$$\mathbb{P}\left[\left|\sum_{i=1}^{t} a_i\right| \geq c^2(t+1)L^2\eta^2\log(dT/\zeta)\middle|\|x_i - y_i\| \leq c\log(dT/\zeta)\eta L\sqrt{i}, \forall i \leq t\right] \leq 1 - \zeta/T.$$

Hence we know

$$\mathbb{P}\left[\|x_{t+1} - y_{t+1}\|^2 \geq c^2\log(dT/\zeta)\eta^2 L^2(t+1)\middle|\|x_i - y_i\| \leq c\log(dT/\zeta)\eta L\sqrt{i}, \forall i \leq t\right] \leq 1 - \zeta/T.$$

Combining the above pieces completes the inductive step, showing that $\|x_{t+1} - y_{t+1}\| \leq c\sqrt{(t+1)\log(dT/\zeta)}\eta L + 2\eta L$ with probability at least $1 - (t+1)\zeta/T$. This completes the proof.

$\square$

By combining Lemmas 2.3 and B.3 with the localization proof technique of [FKT20], we can now prove the excess risk guarantee of Theorem B.1:

*Proof of Theorem 2.1 - Excess risk.* **Excess Risk:** First, we will argue that $\tilde{x}_i = \frac{1}{C}\sum_{j=1}^{C} \tilde{x}_{i,j}$ for all $i$ with high probability $\geq 1 - 3/nm$. Lemma 2.3 implies that

$$\|\tilde{x}_{i,j} - \tilde{x}_{i,j'}\| \leq \tau_i$$

for all $i \in [l]$, $j, j' \in [C]$ with probability at least $1 - 1/nm$. Thus, $s_i(\tau_i) = C$ with probability at least $1 - 1/nm$. Now, conditional on $s_i(\tau_i) = C$, we have $\hat{s}_i(\tau_i) \geq 4C/5$ for all $i$ with probability at least $1 - 1/nm$ by Laplace concentration and a union bound. Moreover, if $\|\tilde{x}_{i,j} - \tilde{x}_{i,j'}\| \leq \tau_i$ for all $j, j'$, then $p_{i,j} = 1$ for all $j$ and hence there are no outliers: $\mathcal{S}_i = \{\tilde{x}_{i,j}\}_{j\in[C]}$. By a union bound, we conclude that $\mathcal{S}_i = \{\tilde{x}_{i,j}\}_{j\in[C]}$ and hence $\tilde{x}_i = \frac{1}{\mathcal{S}_i}\sum_{D_{i,j}\in\mathcal{S}_i} \tilde{x}_{i,j}$ for all $i$ with probability at least $\geq 1 - 3/nm$. By the law of total expectation and Lipschitz continuity, it suffices to condition on this high probability good event that $\tilde{x}_i = \frac{1}{C}\sum_{j=1}^{C} \tilde{x}_{i,j}$ for all $i$: the total expected excess risk can only be larger than the conditional excess risk by an additive factor of at most $3LR/nm$.

Now, conditional on $\tilde{x}_i = \frac{1}{\mathcal{S}_i}\sum_{D_{i,j}\in\mathcal{S}_i} \tilde{x}_{i,j}$ , Lemma B.3 and Jensen's inequality implies

$$\mathbb{E}[F(\tilde{x}_i) - F(\tilde{x}_{i-1})] \lesssim \frac{\mathbb{E}\|\tilde{x}_{i-1} - x_{i-1}\|^2}{\eta_i T_i} + \eta_i L^2 = \frac{d\sigma_{i-1}^2}{\eta_i T_i} + \eta_i L^2. \tag{7}$$

Next, let $x_0^* := x^* = \text{argmin}_{x\in\mathcal{X}} F(x)$, and write

$$\mathbb{E}[F(x_l) - F^*] = \sum_{i=1}^{l} \mathbb{E}[F(x_i^*) - F(x_{i-1}^*)] + \mathbb{E}[F(x_l) - F(x_l^*)]$$

$$\lesssim \frac{R^2}{\eta T_1} + \eta L^2 + \sum_{i=2}^{l} \left[\eta_{i-1}L^2 d + \eta_i L^2\right] + L^2\sqrt{d}\eta_l\sqrt{T_l}$$

$$\lesssim \frac{R^2}{\eta T_1} + d\eta L^2 + L^2\sqrt{d}\eta_l\sqrt{T_l}.$$

Plugging in the prescribed algorithmic parameters completes the excess risk proof. $\square$

## C   Proofs of Results in Section 3

**Theorem C.1** (Re-statement of Theorem 3.1). *Let* $\varepsilon \leq 10$, $q > 0$ *such that* $n^{1-q} > \frac{100\log(20nmde^\varepsilon/\delta)}{\varepsilon(1-(1/2)^q)}$. *Then, Algorithm 3 is* $(\varepsilon, \delta)$-*DP.*

We require the following lemma, which is a direct consequence of [AL24, Lemma 3.5]:

**Lemma C.2.** *Consider Algorithm 2. Let $\mathcal{D}'_i$ and $\mathcal{D}'_i$ be two data sets that differ in the data of one user. Let $E_i = \{|v_i^t| \leq K_i/20 \; \forall t \in [T_i] \cap |\xi_i| \leq K_i/20\}$. Define $E'_i$ similarly for independent draws of random Laplace noise: $E'_i = \{|(v_i^t)'| \leq K_i/20 \; \forall t \in [T_i] \cap |\xi'_i| \leq K_i/20\}$. Let $a_i^t = \mathbf{1}(\widehat{s}_i^t(\mathcal{D}_i) \geq 4K_i/5)$ and $b_i^t = \mathbf{1}(\widehat{s}_i^t(\mathcal{D}'_i) \geq 4K_i/5)$ denote the concentration scores in iteration $t$. Then, conditional on $E_i \bigcap E'_i$ and conditional on $a_i^t = b_i^t$, there exists a coupling $\Gamma_i^t$ over $g_i^t(\mathcal{D}_i)$ and $g_i^t(\mathcal{D}'_i)$ such that for $(h, h')$ drawn from $\Gamma_i$, we have*

$$\|h - h'\| \lesssim \frac{\tau \log(1/\zeta)}{K_i}$$

*with probability at least $1 - \zeta$.*

*Proof of Theorem C.1.* Note that our assumption on $n^{1-q}$ being sufficiently large implies that $n_i \gtrsim \frac{\log(nmd/\delta)}{\varepsilon}$ for all $i \in [l]$. By parallel composition [McS09] and post-processing, it suffices to show that $\{\widehat{g}_i^t\}_{t=1}^{T_i}$ satisfies $(\varepsilon, \delta)$-user-level DP for any $i \in [l]$. To that end, fix any $i \in [l]$ and let $\mathcal{D}$ and $\mathcal{D}'$ be adjacent datasets that differ in the data of one user such that $\mathcal{D}_i \neq \mathcal{D}'_i$. We will show that $\{\widehat{g}_i^t(\mathcal{D})\}_{t=1}^{T_i}$ and $\{\widehat{g}_i^t(\mathcal{D}')\}_{t=1}^{T_i}$ are $(\varepsilon, \delta)$-indistinguishable, which will imply that Algorithm 3 is $(\varepsilon, \delta)$-user-level DP.

Let $E_i = \{|v_i^t| \leq K_i/20 \; \forall t \in [T_i] \cap |\xi_i| \leq K_i/20\}$. Define $E'_i$ similarly for independent draws of random Laplace noise: $E'_i = \{|(v_i^t)'| \leq K_i/20 \; \forall t \in [T_i] \cap |\xi'_i| \leq K_i/20\}$. Our choice of $K_i \geq \frac{500 \log(nmde^\varepsilon/\delta)}{\varepsilon}$ ensures that

$$\mathbb{P}\left(E_i \bigcap E'_i\right) \geq 1 - \delta/(10e^\varepsilon),$$

by Laplace concentration and a union bound. Let $\zeta := \delta/(10 T_i e^\varepsilon)$.

Let $a_i^t = \mathbf{1}(\widehat{s}_i^t(\mathcal{D}) \geq 4K_i/5)$ and $b_i^t = \mathbf{1}(\widehat{s}_i^t(\mathcal{D}') \geq 4K_i/5)$. Note that if $a_i^t = b_i^t = 0$, then $\widehat{g}_i^t(\mathcal{D}) = 0 = \widehat{g}_i^t(\mathcal{D}')$.

Conditional on the good event that $a_i^t = b_i^t$ for all $t$ and conditional on $E_i \bigcap E'_i$, we can bound the $\ell_2$-sensitivity of $g_i^t$ with high probability, via Lemma C.2 and a union bound:

$$\|g_i^t(\mathcal{D}) - g_i^t(\mathcal{D}')\| \lesssim \frac{\tau \log(1/\zeta)}{K_i} \lesssim \frac{\tau \log(nme^\varepsilon/\delta)}{K_i} \tag{8}$$

for all $t \in [T_i]$ with probability at least $1 - T_i \zeta = 1 - \delta/(10e^\varepsilon)$.

Note that $\{\widehat{s}_i^t(\mathcal{D})\}_{t=1}^{T_i}$ and $\{\widehat{s}_i^t(\mathcal{D}')\}$ are $\varepsilon/2$-indistinguishable by the DP guarantees of AboveThreshold in Lemma A.9, since the sensitivity of $s_i^t$ is upper bounded by 2. Therefore,

$$\mathbb{P}(\{a_i^t\}_{t=1}^{T_i} = v, E_i) \leq e^{\varepsilon/2}\left[\mathbb{P}(\{b_i^t\}_{t=1}^{T_i} = v, E'_i) + \zeta\right] \tag{9}$$

for any $v \in \{0,1\}^{T_i}$.

Now, by the sensitivity bound (8), the privacy guarantee of the Gaussian mechanism (Lemma A.6) and our choice of $\sigma_i$, the advanced composition theorem (Lemma A.7), and privacy amplification by subsampling (Lemma A.8), we have

$$\mathbb{P}(\{\widehat{g}_i^t(\mathcal{D})\}_{t=1}^{T_i} \in \mathcal{O} \mid E_i \{a_i^t\}_{t=1}^{T_i} = v) \leq e^{\varepsilon/2}\mathbb{P}(\{\widehat{g}_i^t(\mathcal{D}')\}_{t=1}^{T_i} \in \mathcal{O} \mid E'_i, \{b_i^t\}_{t=1}^{T_i} = v) + (T_i + 1)\zeta, \tag{10}$$

for any event $\mathcal{O} \subset \mathcal{X}$. Here we also used the fact that $K_i \geq \frac{n_i \varepsilon}{\sqrt{T_i}}$.

For short-hand, write $\{a_i^t\}_{t=1}^{T_i} = 1$ if $a_i^t = 1$ for all $t \in [T_i]$ and $\{a_i^t\}_{t=1}^{T_i} = 0$ if $a_i^t = 0$ for some $t \in [T_i]$; similarly for $b_i^t$. Then since the algorithm halts and returns $\{\widehat{g}_i^t(\mathcal{D})\}_{t=1}^{T_i} = 0$ if $\{a_i^t\}_{t=1}^{T_i} = 0$, we know that

$$\mathbb{P}(\{\widehat{g}_i^t(\mathcal{D})\}_{t=1}^{T_i} \in \mathcal{O} \mid E_i, \{a_i^t\}_{t=1}^{T_i} = 0) = \mathbb{P}(\{\widehat{g}_i^t(\mathcal{D}')\}_{t=1}^{T_i} \in \mathcal{O} \mid E'_i, \{b_i^t\}_{t=1}^{T_i} = 0), \tag{11}$$

for any event $\mathcal{O} \subset \mathcal{X}$.

Combining the above pieces, we have

$$\mathbb{P}(\{\widehat{g}_i^t(\mathcal{D})\}_{t=1}^{T_i} \in \mathcal{O}) = \mathbb{P}(\{\widehat{g}_i^t(\mathcal{D})\}_{t=1}^{T_i} \in \mathcal{O} \mid E_i)\mathbb{P}(E_i) + \mathbb{P}(\{\widehat{g}_i^t(\mathcal{D})\}_{t=1}^{T_i} \in \mathcal{O} \mid E_i^c)\mathbb{P}(E_i^c)$$

$$\leq \mathbb{P}(\{\widehat{g}_i^t(\mathcal{D})\}_{t=1}^{T_i} \in \mathcal{O} \mid E_i, \{a_i^t\}_{t=1}^{T_i} = 1)\mathbb{P}(E_i, \{a_i^t\}_{t=1}^{T_i} = 1)$$
$$+ \mathbb{P}(\{\widehat{g}_i^t(\mathcal{D})\}_{t=1}^{T_i} \in \mathcal{O} \mid E_i, \{a_i^t\}_{t=1}^{T_i} = 0)\mathbb{P}(E_i, \{a_i^t\}_{t=1}^{T_i} = 0) + 2T\zeta$$
$$\overset{(i)}{\leq} e^{\varepsilon/2}\mathbb{P}(\{\widehat{g}_i^t(\mathcal{D}')\}_{t=1}^{T_i} \in \mathcal{O} \mid E_i', \{b_i^t\}_{t=1}^{T_i} = 1)e^{\varepsilon/4}\left[\mathbb{P}(E_i', \{b_i^t\}_{t=1}^{T_i} = 1) + T\zeta\right]$$
$$+ \mathbb{P}(\{\widehat{g}_i^t(\mathcal{D}')\}_{t=1}^{T_i} \in \mathcal{O} \mid E_i', \{b_i^t\}_{t=1}^{T_i} = 0)e^{\varepsilon/2}\left[\mathbb{P}(E_i', \{b_i^t\}_{t=1}^{T_i} = 0) + T\zeta\right] + T\zeta$$
$$\leq e^{\varepsilon/2}\mathbb{P}(\{\widehat{g}_i^t(\mathcal{D}')\}_{t=1}^{T_i} \in \mathcal{O}, E_i') + 4T\zeta\left(2e^{\varepsilon/2} + 1\right)$$
$$\leq e^{\varepsilon}\mathbb{P}(\{\widehat{g}_i^t(\mathcal{D}')\}_{t=1}^{T_i} \in \mathcal{O}) + \delta,$$

where $(i)$ follows from inequalities (9), (10), and (11). Thus, $\{\widehat{g}_i^t(\mathcal{D})\}_{t=1}^{T_i}$ is $(\varepsilon, \delta)$-user-level-DP, which implies the result.

$\square$

**Theorem C.3** (Formal statement of Theorem 3.2). *Let $\varepsilon \leq 10$ and $\delta < 1/(mn)$. Then, choosing $\lambda = \frac{L}{R}\left(\frac{1}{\sqrt{nm}} + \frac{\sqrt{d}}{\varepsilon n \sqrt{m}}\right)$ and $p \geq 3q + 2.5 + \log_n(\sqrt{m})$ in Algorithm 3 yields optimal excess risk:*

$$\mathbb{E}F(x_l) - F^* \leq LR \cdot \widetilde{O}\left(\frac{1}{\sqrt{mn}} + \frac{\sqrt{d\log(1/\delta)}}{\varepsilon n \sqrt{m}}\right).$$

*The gradient complexity of this algorithm is upper bounded by*

$$mn\left(1 + \varepsilon\left(\frac{\beta R}{L}\right)^{1/4}\left((mn)^{1/8} \wedge \left(\frac{\varepsilon^2 n^2 m}{d}\right)^{1/8}\right)\right) + \sqrt{\frac{\beta R}{L}}\left(\frac{n^{1/4}m^{5/4}}{\varepsilon} + \left(\frac{n^{1/2}m^{5/4}}{d^{1/4}\varepsilon^{1/2}}\right)\right).$$

We will need the following bound on the excess empirical risk of accelerated (noisy) SGD for the proof of Theorem C.3:

**Lemma C.4.** *[GL13, Proposition 7] Let $x^T$ be computed by $T$ steps of (multi-stage) Accelerated Minibatch SGD on $\lambda$-strongly convex, $\beta$-smooth $\widehat{F}$ with unbiased stochastic gradient estimator $g^t$ such that $\mathbb{E}\|g^t - \nabla\widehat{F}(x^t)\|^2 \leq V^2$ for all $t \in [T]$. Then,*

$$\mathbb{E}[\widehat{F}(x^T) - \min_{x \in \mathcal{X}}\widehat{F}(x)] \lesssim [\widehat{F}(x^0) - \min_{x \in \mathcal{X}}\widehat{F}(x)]\exp\left(-T\sqrt{\frac{\lambda}{\beta}}\right) + \frac{V^2}{\lambda T}.$$

*Remark* C.5. As noted in Lemma C.4, we technically need to call a *multi-stage implementation* of Algorithm 2 in line 7 of Algorithm 3 (as in [GL13]) to get the desired excess risk bound for minimizing the regularized ERM problem in each iteration. For improved readability, we omitted these details in the main body.

Next, we obtain a bound on the variance of the noisy stochastic minibatch gradient estimator $\widehat{g}_i^t$ in Algorithm 2, which can then be plugged in for $V^2$ in Lemma C.4.

**Lemma C.6** (Re-statement of Lemma 3.5). *Let $\delta \leq 1/(nm), \varepsilon \lesssim 1$. Denote $\widetilde{F}_i(x) := \frac{1}{n_i}\sum_{Z_{i,j} \in D_i}\widehat{F}(x, Z_{i,j})$. Then, conditional on $\mathcal{S}_i^t = D_i^t$ for all $i \in [l], t \in [T_i]$, we have*

$$\mathbb{E}\|g_i^t - \nabla\widetilde{F}_i(x_{i-1}^t)\|^2 \lesssim \frac{L^2\log(ndm)}{Km}$$

*for all $i \in [l], t \in [T_i]$, where the expectation is over both the random i.i.d. draw of $\mathcal{D} = (Z_1, \ldots, Z_n) \sim P^{nm}$ and the randomness in Algorithm 3.*

*Proof.* By [LJCJ17, Lemma A.1], we know that, conditional on the draw of the data $D_i$ and for fixed $x_{i-1}^t$, the variance of the minibatch estimator of the gradient of the empirical loss is

$$\mathbb{E}\left[\left.\left\|g_i^t - \nabla\widetilde{F}_i(x_{i-1}^t)\right\|^2 \right| D_i, x_{i-1}^t\right] = \mathbb{E}_{\{i_l\}_{l=1}^K \sim \text{Unif}([n])}\left[\left.\left\|\frac{1}{Km}\sum_{l=1}^K\sum_{j=1}^m \nabla f(x_{i-1}^t, z_{i_l,j}^t)) - \nabla\widetilde{F}_i(x_{i-1}^t)\right\|^2 \right| D_i, x_{i-1}^t\right]$$

$$\leq \frac{\mathbf{1}(K=n)}{K}\frac{1}{n}\sum_{i=1}^{n}\left\|\frac{1}{m}\sum_{j=1}^{m}[\nabla f(x_{i-1}^{t},z_{i,j}^{t})-\nabla\widetilde{F}_{i}(x_{i-1}^{t})]\right\|^{2}.$$
(12)

Recall $\widetilde{F}_i(x):=\frac{1}{n_i m}\sum_{z\in D_i}f(x,z)$ is the empirical loss of $D_i$.

Now, for any fixed $x$ and any $Z\in D_i$, Hoeffding's inequality implies that

$$\|\nabla\widehat{F}(x,Z)-\nabla F(x)\|\leq\tau=O\left(\frac{L\sqrt{\log(nd/\gamma)}}{\sqrt{m}}\right)$$

with probability at least $1-\gamma$, where $\widehat{F}(x,Z):=\frac{1}{m}\sum_{z\in Z}f(x,z)$ is user $Z$'s empirical loss. Thus, by the stability of user-level DP (see [BS23, Theorem 3.4]), for any $i\in[l],t\in[t]$, we have that

$$\|\nabla\widehat{F}(x_{i-1}^{t},Z)-\nabla F(x_{i-1}^{t})\|\leq\tau$$
(13)

for all $Z\in\mathcal{D}$ with probability at least $1-\gamma'=n(e^{2\varepsilon}\gamma+\delta)$, since $x_{i-1}^{t}$ is $(\varepsilon,\delta)$-user-level DP. To make $\gamma'\lesssim 1/m$, we choose $\gamma=1/mn$ and use the assumptions that $\varepsilon\lesssim 1$ and $\delta\leq 1/mn$. Thus, for any fixed $i,t$ we have

$$\|\nabla\widehat{F}(x_{i-1}^{t},Z)-\nabla F(x_{i-1}^{t})\|\lesssim\frac{L\sqrt{\log(n^{2}md)}}{\sqrt{m}}$$

for all $Z\in\mathcal{D}$ with probability at least $1-1/m$, which implies

$$\mathbb{E}\|\nabla\widehat{F}(x_{i-1}^{t},Z)-\nabla F(x_{i-1}^{t})\|^{2}\lesssim\frac{L^{2}\log(nmd)}{m}.$$

This also implies

$$\mathbb{E}\|\nabla\widehat{F}_{\mathcal{D}}(x_{i-1}^{t})-\nabla F(x_{i-1}^{t})\|^{2}\lesssim\frac{L^{2}\log(nmd)}{m},$$

by Jensen's inequality, where $\widehat{F}_{\mathcal{D}}(x)$ is the empirical loss over the entire data set $\mathcal{D}$.

Plugging the above bounds into (12) then yields

$$\mathbb{E}\|g_{i}^{t}-\nabla\widetilde{F}_{i}(x_{i-1}^{t})\|^{2}\lesssim\frac{L^{2}\log(ndm)}{Km}=\mathbb{E}_{\mathcal{D}\sim P^{nm},\{i_{l}\}_{l=1}^{K}\sim\mathrm{Unif}([n])}\left[\left\|\frac{1}{Km}\sum_{l=1}^{K}\sum_{j=1}^{m}\nabla f(x_{i-1}^{t},z_{i_{l},j}^{t})-\nabla\widehat{F}_{\mathcal{D}}(x_{i-1}^{t})\right\|^{2}\right]$$

$$\lesssim\frac{\mathbf{1}(K=n)}{K}\cdot\frac{L^{2}\log^{2}(nmd)}{m},$$

completing the proof.

$\square$

We are now ready to prove Theorem C.3:

*Proof of Theorem C.3.* **Excess risk:** Note that the assumption in the theorem ensures that $K_i\leq n_i$ for all $i$. By similar arguments to those used in [AL24, Proposition 3.7], we will show that with high probability $\geq 1-2/(nm)$, for all $i\in[l],t\in[T_i], \mathcal{S}_i^t=D_i^t$ and hence $g_i^t$ is an unbiased estimator of $\nabla\widehat{F}_{D_i^t}(x_i^t)$. To show this, first note that for any $\gamma>0$ and any fixed $x$,

$$\|\nabla\widehat{F}(x,Z_j)-\nabla F(x)\|\leq\frac{L\log(nd/\gamma)}{\sqrt{m}}$$

with probability at least $1-\gamma/K_i$ by Hoeffding's inequality (see [AL24, Lemma 4.3]). Next, we invoke the stability of differential privacy to show that for all $t\in[T_i], (q_t(Z_{i,1}^t),\ldots,q_t(Z_{i,K_i}^t))$ is $\tau$-concentrated (i.e. there exists $q^*\in\mathbb{R}^d$ such that $\|q_t(Z_{i,j}^t-q^*\|\leq\tau)$ with probability at least $1-T_i(e^{2\varepsilon}\gamma+\delta)$ (see [BS23, Theorem 4.3]). By a union bound and the choice of

$\gamma = 1/[(nm)^{5/4} \log(ndm)e^{2\varepsilon}]$, we have that $(q_t(Z_{i,1}^t), \ldots, q_t(Z_{i,K_i}^t))$ is $\tau$-concentrated for all $i \in [l], t \in [T_i]$ with probability at least $1 - 1/nm$. Now, $\tau$-concentration of $(q_t(Z_{i,1}^t), \ldots, q_t(Z_{i,K_i}^t))$ implies $s_i^t(\tau) = K_i$. Further, $s_i^t(\tau) = K_i$ implies $\widehat{s}_i^t(\tau) \geq 4K_i/5$ with probability at least $1 - \zeta$ if $K_i \geq 500 \log(nm/\zeta)$, by Laplace concentration and a union bound. Next, note that $\tau$-concentration of $(q_t(Z_{i,1}^t), \ldots, q_t(Z_{i,K_i}^t))$ implies $p_i^t(Z_{i,j}^t) = 1$ for all $j \in [K_i]$ and $\mathcal{S}_i^t = D_i^t$. Thus, $\mathcal{S}_i^t = D_i^t$ for all $i, t$ with probability at least $1 - 1/nm$. Setting $\zeta = 1/nm$ and using a union bound shows that with probability at least $1 - 2/nm$, we have $\mathcal{S}_i^t = D_i^t$ and $g_i^t = \nabla \widehat{F}_{D_i^t}(x_i^t)$ for all $i, t$.

Now, Lemma C.4 implies that, *if the outlier-removal procedure in Algorithm 2 leads to an unbiased gradient estimator $g_i^t$ (line 12) for all $t \in [T_i = \widetilde{\Theta}(1 + \sqrt{\beta/\lambda_i})]$, then*

$$\mathbb{E}[\widehat{F}_i(x_i^{T_i}) - \min_{x \in \mathcal{X}} \widehat{F}_i(x)] \lesssim \frac{V_i^2}{\lambda_i T_i}, \tag{14}$$

where $V_i^2 = \max_{t \in [t_i]} \mathbb{E}\|\widehat{g}_i^t - \nabla \widehat{F}_i(x_i^t)\|^2 \lesssim d\sigma_i^2 + \frac{\log(ndm)L^2}{K_i m}$ (unconditionally, after taking expectation over the random draw of $\mathcal{D} \sim P^{nm}$, by Lemma 3.5). We have shown that the event $\text{GOOD} := \{g_i^t = \nabla \widehat{F}_{D_i^t}(x_i^t) \text{ for all } i \in [l], t \in [T_i]\}$ occurs with probability at least $1 - 2/nm$. We will condition on GOOD for the rest of the proof: note that the Lipschitz assumption implies that the total (unconditional) excess risk will only by larger than the conditional (on GOOD) excess risk by an additive factor of at most $2LR/\sqrt{nm}$.

By stability of regularized ERM (see [SSSSS09]), we have

$$\mathbb{E}[F(x_i^*) - F(y)] \lesssim \frac{L^2}{\lambda_i n_i m} + \lambda_i \mathbb{E}[\|x_{i-1} - y\|^2] \tag{15}$$

for all $i$, where $x_i^* := \text{argmin}_x \widehat{F}_i(x)$. By strong convexity and (14), we have

$$(\lambda_i/2)\mathbb{E}\|x_i - x_i^*\|^2 \leq \mathbb{E}\widehat{F}_i(x_i) - \widehat{F}_i^* \lesssim \frac{d\sigma_i^2}{\lambda_i T_i} + \frac{L^2 \log(ndm)}{\lambda_i T_i K_i m}. \tag{16}$$

Thus,

$$\mathbb{E}\|x_i - x_i^*\|^2 \lesssim \frac{d\sigma_i^2}{\lambda_i T_i} + \frac{L^2 \log(ndm)}{\lambda_i T_i K_i m} \lesssim \frac{d\tau^2 \log(1/\delta)}{\lambda_i^2 \varepsilon^2 n_i^2} + \frac{L^2 \log(ndm)}{\lambda_i^2 T_i K_i m} \tag{17}$$

Now, letting $x_0^* := x^* = \text{argmin}_{x \in \mathcal{X}} F(x)$ and hiding logarithmic factors, we have:

$$\mathbb{E}[F(x_l) - F^*] = \sum_{i=1}^{l} \mathbb{E}[F(x_i^*) - F(x_{i-1}^*)] + \mathbb{E}[F(x_l) - F(x_l^*)]$$

$$\lesssim \frac{L^2}{\lambda_1 n_1 m} + \lambda_1 R^2 + \sum_{i=2}^{l} \mathbb{E}\left[\frac{L^2}{\lambda_i n_i m} + \lambda_i \|x_{i-1} - x_{i-1}^*\|^2\right] + L\mathbb{E}\|x_l - x_l^*\|$$

$$\lesssim \frac{L^2}{\lambda n m} + \lambda R^2 + \sum_{i=2}^{l}\left[\frac{L^2}{\lambda_i n_i m} + \lambda_i\left(\frac{d\tau^2 \log(1/\delta)}{\lambda_{i-1}^2 \varepsilon^2 n_{i-1}^2} + \frac{L^2}{\lambda_{i-1}^2 T_{i-1} K_{i-1} m}\right)\right]$$

$$+ L\frac{\sqrt{d}\tau\sqrt{T_l \log(1/\delta)}}{\lambda_l \varepsilon n_l},$$

where the first inequality used (15) and Lipschitz continuity, the second inequality used (17).

Note that $K_i T_i \geq n_i$. Further, our choice of sufficiently large $p$ makes $\lambda_l$ large enough that $L\frac{\sqrt{d}\tau\sqrt{T_l \log(1/\delta)}}{\lambda_l \varepsilon n_l} \leq \frac{LR\sqrt{d}}{\varepsilon n \sqrt{m}}$. Therefore, upper bounding the sum by it's corresponding geometric series gives us

$$\mathbb{E}[F(x_l) - F^*] \lesssim \frac{LR\sqrt{d}}{\varepsilon n \sqrt{m}} + \frac{L^2}{\lambda}\left(\frac{1}{nm} + \frac{d\tau^2 \log(1/\delta)}{\varepsilon^2 n^2}\right) + \lambda R^2. \tag{18}$$

Plugging in $\lambda$ completes the excess risk proof.

**Gradient Complexity:** The gradient complexity is $\sum_{i=1}^{l} T_i K_i m$. Plugging in the prescribed choices of $T_i$ and $K_i$ completes the proof. □

# D   Details on the non-smooth algorithm and the proof of Theorem 4.1

For any loss function $f(\cdot, z)$, we define the convolution function $f_r(\cdot, z) := f(\cdot, z) * n_r$ where $n_r$ is the uniform density in the $\ell_2$ ball of radius $r$ centered at the origin in $\mathbb{R}^d$. Specifically, $n_r(y) = \frac{\Gamma(\frac{d}{2}+1)}{\pi^{\frac{d}{2}} r^d}$ for $\|y\| \leq r$, and $n_r(y) = 0$ otherwise. For simplicity, we omit the dependence on $z$ in the following Lemma:

**Lemma D.1** (Randomized Smoothing, [YNS12, DBW12]). *For any $r > 0$, let $\mathcal{X}_r := \mathcal{X} + \{x \in \mathbb{R}^d : \|x\| \leq r\}$. If $f$ is convex and $L$-Lipschitz over $\mathcal{X}_r$, then the convolution function $f_r$ has the following properties:*

- $f_r(x) \leq f(x) \leq f_r(x) + Lr$, *for all $x \in \mathcal{X}$.*

- $f_r$ *is $L$-Lipschitz and convex.*

- $f_r$ *is $\frac{L\sqrt{d}}{r}$-smooth.*

- *For random variables $y \sim n_r$, we have $\mathbb{E}_y[\nabla f(x + y)] = \nabla f_r(x)$.*

The following lemma can be easily seen from the proofs of Theorems 3.1 and 3.2:

**Lemma D.2** (Privacy and utility of Algorithm 3 for general $K_i, T_i$). *Let $\varepsilon \leq 10$, $q > 0$ such that $n^{1-q} > \frac{100 \log(20nmde^\varepsilon/\delta)}{\varepsilon(1-(1/2)^q)}$.*

- *If $K_i \gtrsim \frac{n_i \varepsilon}{\sqrt{T_i}} + \frac{\log(nmde^\varepsilon/\delta)}{\varepsilon}$, then Algorithm 3 is $(\varepsilon, \delta)$-user-level DP.*

- *If $T_i K_i \geq n_i$ and $T_i \gtrsim (1 + \sqrt{\beta/\lambda_i}) \log(ndm)$ for all $i$, then Algorithm 3 achieves optimal excess risk.*

**Theorem D.3** (Formal statement of Theorem 4.1). *Let $\varepsilon \leq 10$, $\delta < 1/(mn)$, and $q > 0$ such that $n^{1-q} > \frac{100 \log(20nmde^\varepsilon/\delta)}{\varepsilon(1-(1/2)^q)}$. Suppose that for any $z$, $f(, z)$ is convex and $L$-Lipschitz over $\mathcal{X}_r$ for $\mathcal{X}_r := \mathcal{X} + \{x \in \mathbb{R}^d : \|x\| \leq r\}$ where $r = \frac{\sqrt{d}}{\varepsilon n \sqrt{m}} R$. Then, running Algorithm 3 with functions $\{f_r(x; z)\}_{z \in \mathcal{D}}$ yields optimal excess risk:*

$$\mathbb{E}F(x_l) - F^* \leq LR \cdot \widetilde{O}\left(\frac{1}{\sqrt{mn}} + \frac{\sqrt{d \log(1/\delta)}}{\varepsilon n \sqrt{m}}\right).$$

*The gradient complexity of this algorithm is upper bounded by*

$$mn\left(1 + n^{3/8} m^{1/4} \varepsilon^{1/4}\right).$$

*Proof.* By Lemma D.1 and our choice of $r$, we have $|f_r(x, z) - f(x, z)| \leq Lr = O(LR \frac{\sqrt{d}}{\varepsilon n \sqrt{m}})$. Set $\lambda = \frac{1}{\sqrt{mn}}$. Then we know that

$$\mathbb{E}F(x_l) - F^* \leq \mathbb{E}\left[F_r(x_l) - F_r^*\right] + O(LR \frac{\sqrt{d}}{\varepsilon n \sqrt{m}}).$$

Further, $F_r$ is $\beta$-smooth for $\beta \leq \frac{L}{R} \varepsilon n \sqrt{m}$. Set $T_i = (1 + \sqrt{\beta/\lambda_i}) \log(ndm) = 1 + n_i^{3/4} m^{1/2} \varepsilon^{1/2} \log(ndm)$ and $K_i = \frac{n_i \varepsilon}{\sqrt{T_i}} + \frac{\log(nmde^\varepsilon/\delta)}{\varepsilon}$. Then Lemma D.2 implies that Algorithm 3 is $(\varepsilon, \delta)$ user-level DP, and yields the excess risk bound

$$\mathbb{E}F_r(x_l) - F_r^* \leq LR \cdot \tilde{O}\left(\frac{1}{\sqrt{mn}} + \frac{\sqrt{d \log(1/\delta)}}{\varepsilon n \sqrt{m}}\right),$$

as desired. The number of gradient evaluations is

$$\sum_{i=1}^{l} T_i K_i m \lesssim mn\left(1 + n^{3/8} m^{1/4} \varepsilon^{1/4}\right).$$

This completes the proof. □

# E  Limitations

Our work weakens the assumptions on the smoothness parameter and the number of users that are needed for user-level DP SCO. Nevertheless, our results still require certain assumptions that may not always hold in practice. For example, we assume convexity of the loss function. In deep learning scenarios, this assumption does not hold and our algorithms should not be used. Thus, user-level DP *non-convex* optimization is an important direction for future research [LUW24]. Furthermore, the assumption that the loss function is convex and uniformly Lipschitz continuous may not hold in certain applications, motivating the future study of user-level DP stochastic optimization with heavy tails [LR22, ALT24].

Our algorithms are also faster than the previous state-of-the-art, including a linear-time Algorithm 1 with state-of-the-art excess risk. However, our error-optimal accelerated Algorithm 3 runs in super-linear time. Thus, in certain applications where a linear-time algorithm is needed due to strict computational constraints, Algorithm 1 should be used instead.

# F  Broader Impacts

Our work on differentially private optimization for machine learning advances the field of privacy-preserving ML by developing techniques that protect the privacy of individuals (users) who contribute data. The significance of privacy cannot be overstated, as it is a fundamental right enshrined in various legal systems worldwide. However, the implications of our work extend beyond its intended benefits, and it is essential to consider both potential positive and negative impacts.

**Positive Impacts:**

1. Enhanced Privacy Protections: By incorporating differential privacy into machine learning models, we can provide strong privacy guarantees for individuals, mitigating the risk of personal data being exposed or misused.

2. Ethical Data Utilization: DP ML enables organizations to leverage data while adhering to ethical standards and privacy regulations, fostering trust among users and stakeholders.

3. Broad Applications: The techniques we develop can be applied across diverse domains, including healthcare, finance, and social sciences, where sensitive data is prevalent. This broad applicability can drive innovations while maintaining privacy.

4. Educational Advancement: Our research contributes to the growing body of knowledge in privacy-preserving technologies, serving as a valuable resource for future studies and fostering an environment of continuous improvement in privacy practices.

**Potential Negative Impacts:**

1. Misuse by Corporations and Governments: There is a risk that our algorithms could be exploited by entities to justify the unauthorized collection of personal data under the guise of privacy compliance. Vigilant oversight and clear regulatory frameworks are necessary to prevent such abuses.

2. Decreased Model Accuracy: While DP ML provides privacy benefits, it can also lead to reduced model accuracy compared to non-private models. This trade-off might have adverse consequences, such as less accurate medical diagnoses or flawed economic forecasts. For example, an overly optimistic prediction of environmental impacts due to lower accuracy could be misused to weaken environmental protections.

While recognizing the potential for misuse and the trade-offs involved, we firmly believe that the advancement and dissemination of differentially private machine learning algorithms offer a net benefit to society. By addressing privacy concerns head-on and advocating for responsible use, we aim to contribute positively to the field of machine learning and uphold the fundamental right to privacy. Through ongoing research, collaboration, and education, we strive to enhance both the capabilities and ethical foundations of machine learning technologies.

