# OpenReview forum: "Faster Algorithms for User-Level Private Stochastic Convex Optimization"
_NeurIPS.cc/2024/Conference — NeurIPS 2024 poster_

### Official Review · Reviewer_Xb14 · 2024-07-04

**Soundness:** 4
**Presentation:** 4
**Contribution:** 3
**Rating:** 7
**Confidence:** 4

**Summary:**

This paper revisits the user-level private stochastic convex optimization (SCO) problem, where each user can possess multiple data points. The contributions of this paper are in two aspects:

1. They propose a linear-time algorithm that attains the same excess risk as the prior linear-time algorithm, but with a milder assumption on the smooth parameter $\beta$ and does not require the number of users $n$ to depend on dimension $d$, making the algorithm more applicable in practice.
2. They propose an algorithm that achieves optimal excess risk with improved gradient complexity compared to previous work in both smooth and non-smooth settings.

The key insight behind the linear-time algorithm is to remove outlier SGD iterates instead of gradients. To achieve this, they prove a stability argument (Lemma 2.3), the utility then follows by the localization arguments in [FKT20].

The optimal algorithm is inspired by the item-level accelerated phased ERM algorithm of [KLL21], which is then applied to a user-level gradient outlier-removal procedure. The non-smooth case is done by a standard randomized smoothing technique.

**Strengths:**

1. DP SCO is a fundamental problem in private machine learning. In many applications, each user can hold more than one data point. Studying user-level DP SCO is of great importance.
2. The proposed algorithms improve the previous algorithms in various settings, relaxing the assumptions and providing a better run time.
3. The writing of this paper is clear.

**Weaknesses:**

The linear-time algorithm only has a suboptimal excess risk, while the optimal algorithm has a gradient complexity higher than linear.

**Questions:**

In the conclusion, it is stated that whether a linear-time algorithm with optimal risk exists for smooth losses is an open question. What about non-smooth losses? If one has a linear-time algorithm for smooth losses, does the randomized smoothing give a linear-time algorithm for non-smooth losses?

**Limitations:**

Successfully discussed.

---

> ### Author Rebuttal · Authors · 2024-08-07
>
> Thank you very much for your thoughtful feedback and positive assessment of our work. We respond to your comments below.
>
> >*In the conclusion, it is stated that whether a linear-time algorithm with optimal risk exists for smooth losses is an open question. What about non-smooth losses?*
>
> Good question. The non-smooth case is much harder, so we propose that future work should first focus on fully solving the smooth case. Even in the easier item-level setting, there are no known optimal and linear-time algorithms for the non-smooth case (in general arbitrary parameter regimes).
>
>
> >*If one has a linear-time algorithm for smooth losses, does the randomized smoothing give a linear-time algorithm for non-smooth losses?*
>
> Unfortunately, no: Randomized smoothing increases the runtime of the algorithm.

---

> > ### Comment · Reviewer_Xb14 · 2024-08-11
> >
> > Thank you for your response.

---

### Official Review · Reviewer_jGa8 · 2024-07-09

**Soundness:** 2
**Presentation:** 2
**Contribution:** 3
**Rating:** 4
**Confidence:** 3

**Summary:**

This paper proposes new mechanisms for private SCO with user-level DP.  Their approach extends prior work on this topic, and reduces the gradient complexity while attaining optimal excess risk.  Three algorithms are proposed (1) a linear time algorithm that pushes state of the art (but does not achieve optimal excess risk), (2) a (m n)^{9/8} time algorithm that does achieve optimal excess risk (improving over prior work which required (m n)^(3 / 2) gradient evaluation, and (3) an optimal algorithm for non-smooth losses that requires < (m n)^2 gradient computations.  Extensive theoretical results give expressions for the expected excess risk under various assumptions for each mechanism.

**Strengths:**

* The paper improves state of the art on an important problem, reducing gradient complexity while also relaxing assumptions.
* The theoretical results are strong, and clearly stated, presented, and discussed.
* The work is well-motivated and the authors make a compelling story.

**Weaknesses:**

* One of main claims is not supported by the evidence given.  The work is motivated by "modern ML applications, such as training LLMs..." and it claims that "our algorithm and result is applicable to many practical ML problems."  However, looking at Algorithms 1-2 it is not clear that this statement is true.  I believe modern LLMs are pretty big (> 1B parameters) and typically saturate GPU memory, and Alg 1 requires storing C different settings of those parameters to compute the concentration scores -- is that feasible?  From what I can tell, for reasonable values of n, m, epsilon, and delta, C ~ 500.

* The proposed algorithms seem quite complicated.  Is that complexity necessary?  The paper is missing a description + summarization of simpler approaches and their limitations.  E.g., what about plain-old DP-SGD?  What happens when you try applying optimal item-level DP mechanisms to the user-level setting (with appropriate modifications).  What are those modifications (I can think of a few natural ways to extend them).  It would be nice if your mechanism reduced to a known optimal item-level DP mechanism when m=1.

**Questions:**

* In what ways does your algorithm exploit the user-structure of the problem (the fact that each user holds m items)?

* The noise multiplier seems humongous, am I missing something?  It's hard to believe this algorithm doesn't diverge --- do you have any empirical evidence (even on synthetic data satisfying all your required assumptions) to confirm / augment your theorem statements?  Note I have not checked your proofs at all.

epsilon = 10
delta = 1e-6
step size = 1
n = 10 million
m = 100
d = 1 billion
L = 1
--> C = 475
--> T = ? >= 100
--> tau = 400,000
--> sigma = 250,000

**Limitations:**

Limitations are discussed in the appendix.

---

> ### Author Rebuttal · Authors · 2024-08-07
>
> Thank you very much for your thoughtful feedback and assessment of our work.
>
> First, we would like to kindly remind you that our work focuses on understanding theoretical complexity bounds for a fundamental problem. This is both an important goal in its own right and also lays the foundation for future works focusing on practical aspects of user-level DP SCO. We did not intend to give the impression that our algorithms are ready for large-scale practical deployments yet. For example, we did not optimize for constants in our algorithms (e.g. in $\sigma$ and $C$), since our results are stated in big-O.
>
> We respond to your specific comments below.
>
> >*Evidence for the claim "our algorithm and result is applicable to many practical ML problems."*
>
> Thank you for raising this important point.
>
> We stand by this claim, which refers to the applicability of our theoretical result. Theoretically, our Algorithm 1 is applicable whenever $\beta < \sqrt{nmd\varepsilon}$ and $n \gtrsim \log(n/\delta)/\varepsilon$. Typically, $\varepsilon$ is a small constant in practice and $n$ is quite large so that the second condition is easily satisfied. Moreover, $d$ is often large in practice and the smoothness condition holds (e.g. for linear/logistic regression). **For such problems, our algorithm and result is (theoretically) applicable**. We will revise the wording in the final version of our paper to clarify that we are referring to **theoretical applicability** of our result/algorithm to practical ML problems, in contrast to the result/algorithm of [BS23] which does not apply theoretically to these problems.
>
>
> That being said, you raise a valid point about GPU memory and applicability in practice. An important direction for future research is to develop practical implementations of our algorithms that are well suited for large-scale tasks like training LLMs, e.g. by parallelizing computations and optimizing for constants to allow for smaller $C$.
>
> >*The work is motivated by "modern ML applications, such as training LLMs..."*
>
>  We only mentioned LLMs in order to motivate the importance of user-level DP. The full quote reads: *“in many modern ML applications, such as training LLMs on users’ data in federated learning, each user contributes a large number of training examples [XZ24]. In such scenarios, the privacy protection that item-level DP provides for each user is insufficiently weak.”* **We believe that this claim is evidenced by [XZ24]**. We did not mean to suggest that our current algorithms are ready for training LLMs in practice yet. We apologize for any confusion.
>
>
> >*algorithms seem quite complicated. Is that complexity necessary?*
>
> Great question. We believe that some mechanism for private outlier detection and removal is needed to obtain optimal rates for user-level DP SCO. This necessarily introduces additional complexity compared to the item-level setting, where no outlier removal is needed. That being said, we agree that finding simpler algorithms that achieve the optimal rates is an important problem for future work. We will highlight this in the conclusion of the revision.
>
> >*The paper is missing a description + summarization of simpler approaches and their limitations. E.g., what about plain-old DP-SGD? What happens when you try applying optimal item-level DP mechanisms to the user-level setting (with appropriate modifications). What are those modifications (I can think of a few natural ways to extend them)?*
>
> Thank you for the nice suggestion! We will happily add such a description in the camera-ready version if our paper is accepted.
>
> **Simpler modifications of optimal item-level DP algorithms do not result in optimal user-level DP algorithms.** For example, naively applying group privacy to item-level DP-SGD yields suboptimal excess risk. Simple DP-SGD with outlier removal was essentially the algorithm of [AL24], which our algorithm improves over. Moreover, our results cannot be obtained by applying black-box item-level to user-level conversions (e.g. [Bun et al., STOC 2023]). Please let us know if you have any other simple approaches in mind.
>
> >*In what ways does your algorithm exploit the user-structure of the problem (the fact that each user holds m items)?*
>
> We use the fact that each user holds $m$  items to add less noise than would be possible if each user only held 1 item. This is accomplished through our outlier removal procedures and by arguing that with high probability, users’ gradients (or iterates) concentrate around a ball that shrinks with $m$.
>
> >*The noise multiplier seems humongous, am I missing something? It's hard to believe this algorithm doesn't diverge…I have not checked your proofs at all.*
>
> First, our proofs show that **our algorithms are guaranteed to converge**. Moreover, the convergence rates and runtimes are optimal/state-of-the-art (up to constant and log factors), as we discuss in the paper. Second, **we do not optimize for constants** since this is a theoretical work concerned with asymptotic complexity bounds. Thus, it is very likely that a smaller noise multiplier can be used. We leave it for future work to implement practical versions of our algorithms with carefully optimized constants.
>
> >*Do you have any empirical evidence (even on synthetic data satisfying all your required assumptions) to confirm / augment your theorem statements?*
>
> To reiterate, the focus of our paper is on understanding the theoretical complexity bounds for a fundamental problem, not empirical performance. We leave it for future work to implement practical versions of our algorithms and evaluate these algorithms empirically.

---

> > ### Comment · Reviewer_jGa8 · 2024-08-07
> >
> > Acknowledging as a paper that is 100% theoretical, I think it is fair to judge the work on its theoretical rather than practical merits.
> >
> > However, it still seems this approach is a bit more complicated than it ought to be.  For example, taking Alg 1 from [BFTT14](https://arxiv.org/pdf/1908.09970) as a starting point, we could modify it so that in step 5 each batch only contains a single example per user.  Accounting wise, this user-level privacy properties of this algorithm on a dataset with n users is identical to the example-level privacy properties of Alg 1 on a dataset with n examples.  As far as I can tell, Thm 3.2 still goes through, with the sqrt(d log(1/delta)) / (epsilon n) term remaining the same, as sigma depends on the number of users only.  The second term 1/sqrt(n) I believe would change to 1/sqrt(n m).  This bound seems comparable to Thm 2.1.
> >
> > While [BFTT14] was not linear time, I believe a similar modification could be made to an optimal linear-time algorithm like [FKT20](https://arxiv.org/pdf/2005.04763), with some tweaking to hyper-parameters.

---

> > > ### Author Response · Authors · 2024-08-08
> > >
> > > Thank you very much for acknowledging the theoretical nature of our paper and for giving us an opportunity to further clarify the drawbacks of other simple approaches.
> > >
> > > **The user-level DP modification of DP-SGD that you propose does not achieve the SOTA bounds that our algorithms achieve:** In particular, your *proposed algorithm cannot achieve the optimal excess risk of our Algorithm 3*. Moreover, a one-pass (linear-time) implementation of your proposed algorithm *does not achieve the SOTA excess risk in linear time of our Algorithm 1*: Note that each of the error terms in our Theorems 2.1 and 3.2 decrease with $m$, whereas the private optimization error term in your proposed algorithm would not depend on $m$.
> > >
> > > *Why the private optimization error term in your proposed algorithm does not decrease with $m$ (in contrast to our algorithms):* The variance of the additive noise in your proposed user-level DP algorithm does not decrease with $m$, since no outlier-detection/removal procedures are used and hence the worst-case user-level sensitivity of gradients in your algorithm is $\Theta(1)$ . By contrast, our algorithms add less noise scaling with $\tau \approx 1/\sqrt{m}$ by incorporating outlier-detection/removal. This is what enables our algorithms to offer superior performance compared to simple approaches that do not involve outlier-removal. By the same reasoning, **applying your simple modification to FKT20 would also not result in algorithms with the SOTA guarantees that our novel algorithms provide**.
> > >
> > > We sincerely hope that our response clarifies your concerns so that you can increase your score. Please let us know if any other questions remain.

---

> > > > ### Comment · Reviewer_jGa8 · 2024-08-08
> > > >
> > > > * Agreed that what I proposed does not match your guarantee for Alg 3.  But it is better than your utility guarantee for Alg 1 as far as I can tell, assuming n > m, since sqrt(m n) < n.  I think it is reasonable to assume n >> m.
> > > >
> > > > * Another simple way I believe you could get a sqrt(m) dependence in term 1 would be to use an existing optimal item-level DP mechanism, and each time you sample a user, you compute the average gradient over all of their examples.  Under your I.i.d. assumption, the variance of the average of m samples would be 1/m smaller.  I believe you should be able to leverage this to add sqrt(m) less noise and recover the rates you get for Alg 3, with a much simpler approach.
> > > >
> > > > I will revisit my evaluation and consider updating my score after your next response.

---

> ### Author Response · Authors · 2024-08-08
>
> Thank you very much for your timely reply.
>
> As a general comment, we emphasize that there is a long line of work on the user-level DP-SCO problem, which tries to get optimal rates via complicated algorithms. Reducing user-level DP to item-level DP, similar to your proposal, is trivial, but does not benefit significantly from larger $m$. The challenge of user-level DP SCO is designing algorithms that benefit (in all of the error terms) from large $m$, as our algorithms do. Indeed, the regime $m > n$ is more interesting theoretically for user-level DP SCO. We will discuss this and the other naive baselines (e.g. group privacy) in the final version.
>
> We respond to your specific comments below.
>
> >*it is better than your utility guarantee for Alg 1 as far as I can tell, assuming n > m...*
>
> First, note that the **runtime of your proposed *suboptimal* algorithm** (using BFTT19) **would need to be at least *quadratic* in $n$** in order to achieve the suboptimal risk bound stated in your above comment, even for smooth losses (after optimizing for $T$). This runtime is worse than any of the algorithms we provide in our paper. In particular, **our Algorithm 3 achieves *optimal* excess risk with runtime that is *subquadratic* in $n$**, scaling with $n^{9/8}$. Moreover, **our Algorithm 1 is linear-time**. Furthermore, a linear-time implementation of your proposed BFTT19-based algorithm (by simply choosing $T$ to be small) would result in a fairly strict restriction on the smoothness parameter ($\beta \lesssim \sqrt{m}$) in order to achieve inferior excess risk of $\approx 1/\sqrt{m} + \sqrt{d}/n\varepsilon$.
>
> Finally, if you try to modify FKT20 with your proposed simple approach, you obtain a suboptimal linear-time algorithm that suffers from a very severe smoothness restriction.
>
>
> All that being said, we agree that simplicity is a virtue and will gladly add discussion of your proposed simple approach to the final version. Thank you very much for your interesting suggestion.
>
>
> >*Under your I.i.d. assumption, the variance of the average of m samples would be 1/m smaller. I believe you should be able to leverage this to add sqrt(m) less noise and recover the rates you get for Alg 3, with a much simpler approach.*
>
>
> Notice that **your proposed approach would fail to satisfy user-level DP** without incorporating some outlier removal step or adding excessive noise (leading to suboptimal risk). This is because *one cannot assume the data is i.i.d. when bounding sensitivity/proving privacy*. That is why we need to use outlier removal in our algorithms.
>
> We really appreciate you engaging with us in this productive discussion and believe that implementing your suggestions will strengthen the final version. Please let us know if you have any further questions or comments.

---

> ### Comment · Reviewer_jGa8 · 2024-08-09
>
> * Reducing user-level DP to item-level DP, similar to your proposal, is trivial, but does not benefit significantly from larger m.
> >* This may be true, although you have not convinced me yet, but either way this discussion is missing in this paper.  You need to make a convincing case of this in the paper, and such changes would be large enough to warrant a fresh review in my opinion.
>
> *  Indeed, the regime is more interesting theoretically for user-level DP SCO.
> >* One of the things I look for based on the reviewer guidelines is "are the claims made supported by evidence."  What evidence do you have to support this statement?  What does it even mean to be "more interesting theoretically"?
>
> * runtime of your proposed **suboptimal** algorithm
> >* I don't believe you adequetely showed this was suboptimal, other than saying (without evidence) that "m > n" is the more interesting regime.  At minimum, you should qualify your claims of optimality with the caveat "under some conditions".
>
> * would need to be at least **quadratic in n**
> >* Again, you are making these claims without evidence, and as far as I can tell is simply not true either.  BFTT sets the batch sizes to n sqrt(epsilon / 4T) and runs for T <= n/8 iterations.  This means that it requires sqrt(epsilon * n) / m passes over the dataset.  If m > n as you purport, this is actually sub-linear, not quadratic.
> >* Second, BFTT14 is 10 years old -- of course there are more recent papers that achieve the same rates with much less runtime.   I did not suggest running BFTT with a small T, but just used that as one of the simplest example algorithm that achieves the optimal rates.  I believe the modifications I proposed are compatible with any optimal item-level DP mechanisms in principle, including the linear-time ones.
>
> * Finally, if you try to modify FKT20 with your proposed simple approach, you obtain a suboptimal linear-time algorithm that suffers from a very severe smoothness restriction.
> >* This very well may be true, I am not familiar enough with the prior results to confirm or deny.  But given the flaws in reasoning I've identified above, I will not trust the statement **without evidence**.
>
> * **your proposed approach would fail to satisfy user-level DP**
> >* I didn't actually propose a specific mechanism here, and maybe it is the case that the only way to get 1/sqrt(m) rates is with outlier removal, but again, you are making this claim without providing evidence.  Can you prove there is no other way to do this?  In principle you should be able to run any linear-time algorithm for item-level DP, and replace the step that does item-level DP mean estimation of the minibatch gradient with a step that does user-level DP mean estimation of the minibatch gradient.  This would lead to a valid algorithm, that leverages and connects to prior work already done on this problem of user-level DP mean estimation.

---

> ### Author Response · Authors · 2024-08-09
>
> First, **there are known trivial approaches that obtain excess risk $1/\sqrt{nm}+\sqrt{d}/n\epsilon$**: e.g. reducing to item-level DP SCO via group privacy or by averaging each user’s gradients. This bound $1/\sqrt{nm}+\sqrt{d}/n\epsilon$ is the same as the excess risk bound that the reviewer claimed their simple approach obtains. Note that while these results are simple to obtain, **they do not benefit significantly from large $m$** because *the privacy term $\sqrt{d}/n\epsilon$ does not shrink as $m$ increases*. Thus, an important challenge in user-level DP SCO is getting risk bounds that decay with large $m$. Getting such bounds is non-trivial and has been the topic of a long line of work (involving complicated algorithms) that we discuss in the Introduction. By contrast, if $m$ is small, then one can simply apply a trivial approach without suffering too much. **For these reasons, the large $m$ regime is more interesting** theoretically.
>
> That being said, we clearly acknowledge that the trivial baseline rate $1/\sqrt{nm}+\sqrt{d}/n\epsilon$ can be better than our linear-time bound in Theorem 2.1 when $n > m$. However, it is not clear to us whether one can obtain the trivial bound both in linear time and with a mild smoothness assumption like the one in our Theorem 2.1. Moreover, we reiterate that **this trivial bound is suboptimal, since it is bigger than the optimal bound given in our Theorem 3.2**. We will add this comparison/discussion in the final revision. Thank you again for this valuable suggestion.
>
> Finally, we want to emphasize again that **we do not see any way for simpler approaches to obtain any of our main results** (Theorems 2.1, 3.2, or 4.1). We believe that developing simpler optimal algorithms for user-level DP SCO is an interesting open question and invite the reviewer to make progress on this problem. We do note that several of the reviewer’s ideas were already explored in early suboptimal works on user-level DP SCO (e.g. LSA+21 and BS23) and therefore will probably not result in optimal rates.
>
> We thank the reviewer once again for their feedback.

---

> > ### Comment · Reviewer_jGa8 · 2024-08-12
> >
> > Acknowledging response.  I think the paper could be significantly stronger by thinking more carefully about these things and incorporating this discussion into the paper.  I am keeping my original rating as is since I think it would be a net positive for the scientific community for this work to be polished a bit more.  I am reducing the soundness rating since I think there are some improvements you can make to the scientific methodology that came up in the discussion.  However, there are some technical merits to the paper, and if other reviewers are happy to accept it as-is, I would not object to acceptance.

---

### Official Review · Reviewer_W7xk · 2024-07-10

**Soundness:** 3
**Presentation:** 3
**Contribution:** 3
**Rating:** 7
**Confidence:** 3

**Summary:**

This paper considers stochastic convex optimization under user-level differential privacy.

Algorithm 1 achieves the previous state-of-the-art excess risk of the linear-time user-level DP algorithm with milder assumptions. The algorithm is based on item-level DP-SGD algorithms. Previous paper AL24 shows that when the data of one user changes, the empirical gradient will not change too much, hence they remove outlier gradients to ensure privacy. Instead of removing outlier gradients, this paper removes outlier SGD iterates. This technique relies on a novel stability bound of SGD iterates proved in this paper.

Algorithm 3 improves the run-time for both beta-smooth and non-smooth loss functions. It applies outlier-gradient removal to random mini-batches and implements previous subroutines to reach a good performance.

**Strengths:**

This paper makes concrete improvements from the previous work. Algorithm 3 has a faster run-time under milder assumptions while achieving optimal excess risk and user-level DP. Algorithm 1 improves the previous state-of-the-art excess risk of the linear-time user-level DP algorithm. Novel techniques and analysis are put forward which may be of independent interest.

**Weaknesses:**

Section 1.1 may contain more information about the intuition of the novel techniques (e.g., what may be the key reason that outlier-iterate removal is better than outlier-gradient removal?).
Contribution 2 of this paper lacks a comparison table to previous results.

**Questions:**

The questions are asked in the 'Weaknesses' part.

---

> ### Author Rebuttal · Authors · 2024-08-07
>
> Thank you very much for your thoughtful feedback and your positive assessment of our work. We respond to your comments below.
>
>
> >*Section 1.1 may contain more information about the intuition of the novel techniques (e.g., what may be the key reason that outlier-iterate removal is better than outlier-gradient removal?)*
>
> Great suggestion. We will elaborate on the intuition behind our techniques in the final version.
>
> Since the item-level DP phased SGD algorithm [FKT20] adds noise to the iterates, removing outlier iterates is a natural approach for extending this algorithm to be user-level DP. If we instead attempt to remove outlier gradients, one issue is that large batch sizes lead to a severe restriction on the smoothness parameter.
>
>
> >*Contribution 2 of this paper lacks a comparison table to previous results.*
>
> This is a very good suggestion. The only previous linear-time algorithm for user-level DP SCO is due to [BS23]. We compare our result against [BS23] in lines 80-93 and Remark 2.2. We will put this comparison into a second table in the final version.

---

> > ### Comment · Reviewer_W7xk · 2024-08-14
> >
> > Thank you for your response! You have answered my questions thoroughly.

---

### Official Review · Reviewer_JSLp · 2024-07-12

**Soundness:** 4
**Presentation:** 3
**Contribution:** 3
**Rating:** 4
**Confidence:** 4

**Summary:**

This paper proposes new algorithms for stochastic convex optimization under user level differential privacy. This paper improves the computation complexity. The first algorithm achieves linear time complexity with suboptimal risk bound (the risk is SOTA among all linear time algorithms). The second algorithm achieves optimal risk with suboptimal time complexity (the time complexity is SOTA among all risk-optimal algorithms).

**Strengths:**

The authors have addressed an important problem. The method improves over Bassily et al. ICML 2023 and Liu et al. AISTATS 2024 on the time complexity. Moreover, this paper removes the assumption n\geq \sqrt{d}. I think that this paper makes a solid and interesting contribution.

**Weaknesses:**

1. I feel that the second algorithm (Section 3 in the paper) is hard to follow. I do not understand the main ideas of the design.

2. Based on my own understanding, it seems that this paper requires $\epsilon<1$, since it divides users into $1/\epsilon$ groups. When $\epsilon>1$, there is only one group. Following the remainder of this paper, I feel that the bound with $\epsilon>1$ may not be optimal. Please correct me if it is wrong.

3.In item-level case, optimal rates with linear time has been achieved in

Feldman et al. Private stochastic convex optimization: optimal rates in linear time. STOC 2020.

Although this paper has cited the above paper, it would be better if the authors can provide more explanations on why achieving linear time with optimal rik is hard for user-level cases. In other words, why does the methods in Feldman et al. can not be simply extended to user-level case. My feeling is that using something like two-stage approach (such as Levy et al. Learning with user level privacy. NeurIPS 2021), the item-level methods can be converted to user-level ones with optimal rate.


Minor issue: The statements of three algorithms are not with the same font. The algorithm 2 is different with algorithm 1 and 3.

In general, I think that the paper indeed makes a solid contribution. However, the writing needs to be further improved and details need to be further polished.

**Questions:**

1. Can the authors provide more intuitive explanations of the design of algorithms, especially the second one?

2. I wonder what will be the risk bound for strong convex loss functions. The case with strong convex loss function has been discussed in

Kamath et al. Improved rates for differentially private stochastic convex optimization with heavy-tailed data. ICML 2022.

This holds for item-level case. I wonder if one can derive the user-level counterparts.

3. Can the authors please provide more discussions on why it is hard to achieve linear time with optimal risk?

**Limitations:**

The authors have addressed limitations well in the conclusion.

---

> ### Author Rebuttal · Authors · 2024-08-07
>
> Thank you very much for your thoughtful feedback and your positive assessment of our work. We respond to your comments below.
>
>
> >*Main ideas of the second algorithm (Section 3 in the paper)*
>
> Our second algorithm is **inspired by the item-level accelerated phased ERM algorithm of [KLL21]**. Their algorithm runs in $\log_2 (n)$ phases. In each phase $i$, a disjoint batch of $n_i$ samples is used to define a regularized empirical loss function, $\hat{F}_i(x)$. The main ideas of the algorithm are:
>
> - A noisy DP variation of **accelerated** SGD is used to efficiently find an approximate DP minimizer $x_i$ of $\hat{F}_i(x)$. This approximate minimizer $x_i$ is then used as the initial point in phase $i+1$.
>
> - By **stability** of regularized ERM, $x_i$ is also an approximate minimizer of the underlying population loss function $F$.
>
> - **Iterative localization**: As $i$ increases and $x_i$ gets closer to  $x^* = argmin_{x} F(x)$ , we increase the regularization parameter (geometrically) to prevent $x_i$ from moving too far away from $x_{i-1}$ and hence from $x^*$. We also shrink $n_i$ geometrically.
>
> Our algorithm is a **user-level DP variation of the algorithm described above**: Our **Algorithm 2 uses outlier-detection and outlier-removal procedures to get a low-variance noisy *user-level DP* estimate of the gradient** (with sensitivity scaling with $\tau \approx 1/\sqrt{m}$) in each iteration. This noisy user-level DP gradient is then used to take a step of accelerated SGD. Algorithm 3 uses Algorithm 2 as a subroutine to get a **stable**, user-level DP approximate solution of a regularized ERM problem in each phase, and applies **iterative localization**.
>
> Please see **Section 1.1 and lines 219-225** for a more detailed description of our algorithmic techniques.
>
> >*it seems that this paper requires $\varepsilon <1$, since it divides users into $1/\varepsilon$ groups…I feel that the bound with $\varepsilon >1$ may not be optimal. Please correct me if it is wrong.*
>
>
> Good question. In fact, **we do not require  $\varepsilon <1$**.
>
> **For Algorithm 1, we do not necessarily need $\varepsilon < 1$:** it suffices that $\varepsilon < 50 \log(20 n m e^{\varepsilon}/\delta)$, so that  $C \geq 2$. This condition is not practically restrictive.
>
> **Algorithm 3 is also optimal for $\varepsilon > 1$:** This algorithm does not divide users into groups that depend on $\varepsilon$.
>
> >*More explanations on why achieving linear time with optimal risk is hard for user-level cases…why the methods in Feldman et al. can not be simply extended to user-level case?*
>
> Excellent question! Note that even obtaining user-level DP algorithms with *polynomial* runtime was challenging and only recently solved by the work of Asi & Liu (2023).
>
> In the item-level DP setting, Feldman et al. give two optimal linear-time algorithms: snowball SGD and phased SGD.
>
> It is not at all clear how to extend snowball SGD into an optimal linear-time user-level DP algorithm.
>
> Therefore, we aimed to extend their phased SGD algorithm into a user-level DP algorithm with our Algorithm 1. The key challenge in obtaining optimal user-level DP excess risk with this algorithm is controlling the **user-level sensitivity of the iterates of one-pass SGD**. In the item-level case, the sensitivity of the iterates is $O(\eta L)$ [Feldman et al., Lemma 4.3], independent of the number of iterations $T$. However, in the user-level case, the sensitivity is $O(\eta L \sqrt{T})$ (our Lemma 2.3), and we believe this bound is tight. Hence the additive Gaussian noise that is needed to privatize the iterates is too large to obtain the optimal rate in the phased SGD framework. Thus, a fundamentally different algorithmic framework that leverages Lemma 2.3 in a more effective way may be needed to obtain the optimal rate in linear time.
>
> A second challenge in designing user-level DP variations of phased SGD is **instability of the outlier-detection scheme**: If the initial point $x_{i-1}$ changes by a small amount and we do outlier-detection, then the output can change greatly. Moreover, outlier detection seems to be necessary for any optimal user-level DP algorithm. Thus, another direction for future work could be to develop a new, **more stable outlier detection method**.
>
> We will add a discussion of these challenges and potential ways to overcome them in the final version of the paper.
>
> >*My feeling is that using something like two-stage approach (such as Levy et al. NeurIPS 2021), the item-level methods can be converted to user-level ones with optimal rate.*
>
> First, recall that the approach of Levy et al. does not obtain optimal rates. Moreover, their two-stage approach is similar to our outlier removal procedure. Their approach  suffers from similar instability as our outlier removal approach. Thus, we do not see any way for their approach to allow for improvements over our algorithms. Please let us know if you have any ideas for how to leverage Levy et al. that we might be missing.
>
> >*What will be the risk bound for strong convex loss functions?*
>
> Great question! The risk bounds for $\mu$-strongly convex loss functions will essentially be **the square of the convex risk bound** (but with the scaling factor $LR$ replaced by $L^2/\mu$). This follows from a reduction in [FKT20]. In particular, **Algorithm 3 can easily be converted into an optimal algorithm for strongly convex functions with state-of-the-art runtime**. We will comment on this in the revision.

---

> > ### Comment · Reviewer_JSLp · 2024-08-08
> > **Further response**
> >
> > Thanks for your reply. Regarding $\epsilon$, in the paper, data are divided into $1/\epsilon$ groups. So if $\epsilon>1$, how to divide it? Do you mean that data are actually divided into $\lceil 1/\epsilon\rceil$ groups? If so, my intuitive feeling is that there should be a phase transition at $\epsilon = 1$. Your bound does not have any phase transition, which looks a bit strange for me. Please correct me if I am wrong.

---

> > > ### Author Response · Authors · 2024-08-08
> > >
> > > Thank you very much for your response and for giving us an opportunity to further clarify this important point.
> > >
> > >
> > > First, to be clear, we do require $\varepsilon$ to be bounded by some constant (e.g. $\varepsilon \leq 100$) in the current algorithm/analysis. Thank you for catching this. We will add this assumption to our statement of Theorem 2.1 in the revision. We believe that this assumption is very reasonable, since the privacy guarantees degrade rapidly as $\varepsilon$ grows.
> > >
> > > Next, we respond to your specific question:
> > >
> > > >*in the paper, data are divided into $1/ϵ$  groups. So if ϵ>1, how to divide it?*
> > >
> > >
> > > In fact, we don’t divide the data into $1/\varepsilon$ groups exactly, but rather we divide the data into $C$ groups, where $C$ is defined in line 2 of Algorithm 1. Note that $C \geq 2$ for any $\varepsilon > 0$.
> > >
> > >
> > > The reason that we choose $C$ the way we do is in order to ensure that the Laplace noise added in line 10 of Algorithm 1 is much smaller than $C$ with high probability. This ensures that the outlier-removal procedure succeeds with high probability.

---

> > > > ### Comment · Reviewer_JSLp · 2024-08-08
> > > > **Reply**
> > > >
> > > > Thanks for your response.

---

> > > > > ### Author Response · Authors · 2024-08-08
> > > > >
> > > > > Thank you very much again for your feedback and for engaging in this productive discussion with us. We hope that our responses clarified your concerns, so that you can re-evaluate your score. If you have any remaining questions, comments, or suggestions for us, please do let us know.

---

> > > > > > ### Comment · Reviewer_JSLp · 2024-08-11
> > > > > >
> > > > > > I have seen your discussion with reviewer jGa8. Your reply is
> > > > > >
> > > > > > "Notice that your proposed approach would fail to satisfy user-level DP without incorporating some outlier removal step or adding excessive noise (leading to suboptimal risk). This is because one cannot assume the data is i.i.d. when bounding sensitivity/proving privacy. That is why we need to use outlier removal in our algorithms."
> > > > > >
> > > > > > This is also not convincing to me. The following two papers discuss the solution of mean estimation under user-level DP:
> > > > > >
> > > > > > [1] Agarwal, Sushant, et al. "Private Mean Estimation with Person-Level Differential Privacy." arXiv:2405.20405.
> > > > > >
> > > > > > [2] Zhao, Puning, et al. "A Huber Loss Minimization Approach to Mean Estimation under User-level Differential Privacy." arXiv:2405.13453.
> > > > > >
> > > > > > Both two papers calculate the user-wise mean first, which reduces the sensitivity to $1/\sqrt{m}$.  User-level DP requirements are satisfied in both papers. Therefore, I can not agree with your statement that the solution proposed by reviewer jGa8 fails to satisfy user-level DP. Moreover, you have mentioned multiple times that simple extensions of item-level methods do not benefit from large $m$. Actually, both methods in [1] and [2] benefit significantly from large $m$. The mean estimation problem is somewhat different from optimization problem. However, a natural solution is to use these mean estimation algorithms to estimate the gradients, and then perform updates for $T$ steps. To bound the risk of optimization, some tricks may be necessary  (such as bounding the generalization gap). Could you please discuss what will the bound be if we just use these user-level mean estimation algorithms to estimate gradients?
> > > > > >
> > > > > > Also, I do not understand "one can not assume the data is i.i.d when bounding sensitivity". As far as I know, existing research on user-level DP usually assume that all samples are i.i.d.
> > > > > >
> > > > > > It is reasonable not to cite these two papers for now because they are both posted after the NeurIPS submission deadline. However, due to these concerns that are also raised by reviewer jGa8, currently I can not increase my overall rating.

---

> ### Author Response · Authors · 2024-08-12
>
> >*A natural solution is to use these mean estimation algorithms to estimate the gradients, and then perform updates for $T$ steps….Could you please discuss what will the bound be if we just use these user-level mean estimation algorithms to estimate gradients?*
>
>
> **The suggested approach of using a user-level DP mean estimator to estimate the gradient in each iteration was taken by [BS23]**. The result of [BS23] is discussed in our paper: see e.g. row 1 of Fig. 1, lines 47-53, and remarks 3.3 & 3.4. Recall that although the [BS23] approach can obtain optimal excess risk for certain sufficiently smooth functions, it has the following limitations: (i) **the requirement on the number of users size is strict: $n \gtrsim d/\epsilon$**; (ii) **strict restriction on the smoothness parameter (see row 1 of Fig. 1) and does not work for non-smooth functions**. By contrast, *our results only require $n \gtrsim \log(d)/\epsilon$, our smoothness requirement is much milder than [BS23], and we get optimal excess risk even for non-smooth* functions.
>
>
> *Technical reasons for limitations (i) and (ii)*: Existing near-optimal $(\epsilon, \delta)$- user-level-DP mean estimators require $n \gtrsim d/\epsilon$ (in the two recent papers the reviewer suggests) or $n\gtrsim \sqrt{d}/\epsilon$ [BS23]. However, if we use full-batch and run DP-GD for $T$ steps, then advanced composition implies that we need $n \gtrsim\sqrt{T d}/\epsilon$. Here, $T$ needs to depend on $n$ to get optimal excess risk, which leads to severe restrictions on the smoothness parameter. In particular, their approach cannot handle non-smooth functions (assuming $\epsilon < \sqrt{d}$) since $T \sim n^2$ would be needed in their algorithm to get optimal risk for such functions.
>
>
> To address the above two limitations, [AL24] designed a new mean estimation sub-procedure based on incorporating outlier removal with the AboveThreshold procedure. Our algorithms build on the techniques of [AL24], as we discussed in Section 1.1.
>
>
> We will include this more detailed discussion/comparison of the [BS23]-type approach and the technical reasons for its limitations in the final version.
>
>
> >*I do not understand "one can not assume the data is i.i.d when bounding sensitivity". As far as I know, existing research on user-level DP usually assume that all samples are i.i.d.*
>
>
> We meant that **one cannot *use* the i.i.d. assumption in the privacy analysis**, because (user-level) DP is a strong *worst-case, distribution-free* requirement that must hold for all pairs of adjacent databases, not just i.i.d. databases.
>
> We thank the reviewer again for giving us an opportunity to further clarify these points.

---

> > ### Comment · Reviewer_JSLp · 2024-08-13
> >
> > Thanks for your further response.
> >
> > **(i) the requirement on the number of users size is strict: $n\gtrsim d/\epsilon$**
> >
> > This is inaccurate. The requirement should be $n\gtrsim \sqrt{d}/\epsilon$ instead of $d/\epsilon$.
> >
> > I agree that your method have a much weaker assumption on the smoothness $\beta$. The risk bound does not depend explicitly on $\beta$. In this aspect, you have solved part of my concerns. However, I still agree with the reviewer jGa8. While I agree that this new method is novel and interesting, this paper needs to be further polished before acceptance. In particular, your discussion with reviewer jGa8 about outlier removal is not fully convincing to me. Therefore, I would still like to keep my score.

---

> ### Author Response · Authors · 2024-08-13
>
> First, we would like to thank the reviewer for replying to our responses in a timely fashion and engaging in this important discussion to clarify the limitations of prior works compared to our work. We respond to your comments below.
>
> >*The requirement should be $n \gtrsim \sqrt{d}/\epsilon$ instead of $d/\epsilon$*
>
> Apologies for the confusion. We meant to write that $n \gtrsim d/\epsilon$ users are needed if one uses the two concurrent papers that the reviewer mentioned [1] and [2]: This can be seen by inspecting Theorem 1.2 in [1] and the discussion that follows the statement of the theorem, and by inspecting Eq. (3) and Theorems 2&3 in [2].
>
> You are right that BS23 "only" needs $n \gtrsim \sqrt{d}/\epsilon$, which we mentioned in the second paragraph of our above response and in line 49 of our paper. Note that this **polynomial dependence on $d$ still leads to a strict requirement on the number of users in comparison to our work, which only requires the number of users to depend logarithmically on $d$**.
>
> >*your discussion with reviewer jGa8 about outlier removal is not fully convincing to me*
>
> Please let us know if there are any specific points in this discussion that you have doubts or questions about. We would be happy to clarify further why we believe that simpler approaches are unable to obtain our SOTA results (without further innovations), and why/how our outlier-removal technique is a powerful way to overcome the technical barriers faced in prior works (e.g. BS23).
>
> >*I agree that your method have a much weaker assumption on the smoothness parameter...and that this new method is novel and interesting*
>
> Thank you for acknowledging these important aspects of our work. We re-iterate that our significant improvements in **runtime** and in the **number of users** needed (logarithmic instead of polynomial dependence on $d$) are also crucial components of our SOTA results.

---

### Author Response · Authors · 2024-08-07
**Note to all Reviewers**

Thank you very much for your thoughtful assessments of our work and your constructive feedback!

We are very happy that *all of the reviewers* appreciated the **strength of our results**, which improve over the state-of-the-art on an “**important problem**” (*Reviewers JSLp, jGa8 and Xb14*). We are also glad that you enjoyed our **writing/presentation**, noting that our “results are strong, and clearly stated, presented, and discussed,” our work is “well-motivated” and makes a “compelling story” (Reviewer jGa8), and that “the writing of this paper is clear” (*Reviewer Xb14*). Additionally, we are pleased that you appreciated our “**Novel techniques and analysis**...which may be of independent interest” (*Reviewer W7xk*).

We respond to each of your comments below. We look forward to receiving your responses to our rebuttals and engaging in a productive discussion.

---

### Decision · Program_Chairs · 2024-09-25

**Decision:**

Accept (poster)

**Comment:**

The paper considers the problem of stochastic convex optimization under user-level privacy and provides efficient algorithms that achieve optimal risk without any assumptions. For non-smooth functions, prior works required (mn)^3 gradient computations and the proposed algorithms require n^{11/8} m^{5/4} computations, where n is the number of users and m is the number of samples per user. For beta-smooth functions, prior works required (mn)^{3/2} gradient computations  and proposed work reduces complexity to (mn)^{9/8} + n^{1/4} m^{5/4}. While these results advance the state of the art, reviewers raise concerns about the complexity of the algorithm and limited intuitive explanations on why this complexity is necessary.  Given the wide-audience of NeurIPS, I strongly encourage authors to add a high-level discussion in the paper to explain the need for each of the steps in the algorithm and how they came up with them.